# Evaluating Ecosystem Characteristics and Ecological Carrying Capacity for Marine Fauna Stock Enhancement Within a Marine Ranching System

**DOI:** 10.3390/ani15020165

**Published:** 2025-01-10

**Authors:** Jie Feng, Haolin Yu, Lingjuan Wu, Chao Yuan, Xiaolong Zhao, Huiying Sun, Cheng Cheng, Yifei Li, Jingyi Sun, Yan Li, Xiaolong Wang, Yongjun Shang, Jiangling Xu, Tao Zhang

**Affiliations:** 1North China Sea Marine Forecasting Center of State Oceanic Administration, Qingdao 266061, China; f823725486@163.com (J.F.); vivioceangk@163.com (L.W.); yuan-chao-hi@163.com (C.Y.); zhaoxiaolong@ncs.mnr.gov.cn (X.Z.); shy0517@126.com (H.S.); hdliyifei@163.com (Y.L.); candybutterfly@126.com (J.S.); 2Key Laboratory of Ecological Prewarning, Protection and Restoration of Bohai Sea, Ministry of Natural Resources, Qingdao 266033, China; 3Shandong Provincial Key Laboratory of Marine Ecological Environment and Disaster Prevention and Mitigation, Qingdao 266061, China; 4Laboratory for Marine Science and Technology, Qingdao National Laboratory for Marine Science and Technology, Qingdao 266237, China; yuhaolin777@outlook.com (H.Y.); jsrgsha@163.com (C.C.); shangyongjun@stu.ouc.edu.cn (Y.S.); 5Center for Ocean Mega-Science, Chinese Academy of Sciences, Qingdao 266400, China; 6CAS Engineering Laboratory for Marine Ranching, Institute of Oceanology, Chinese Academy of Sciences, Qingdao 266400, China; 7CAS Key Laboratory of Marine Ecology and Environmental Sciences, Institute of Oceanology, Chinese Academy of Sciences, Qingdao 266400, China; 8Guangxi Academy of Oceanography, Nanning 530022, China; 2005304001@163.com; 9Marine Science Research Institute of Shandong Province, Qingdao 266104, China; wxl307@163.com

**Keywords:** marine ranching, stock enhancement, index system, carrying capacity, scenarios simulation

## Abstract

Assessing the structure and function of marine ranching ecosystems is crucial for their successful development. In this study, we used energy flow models and an evaluation system to measure the food web structure and ecological carrying capacity of both a marine ranching ecosystem and a nearby control site. Through this, we identified the most suitable species for stock enhancement and proposed strategies to improve the marine ranching ecosystem. We also used a food web model to simulate the effects of these strategies. Our results show that the marine ranching ecosystem was in better condition than the control site, which may confirm the success of the ranching efforts. Mussels, large crabs, and Scorpaenidae were identified as key groups for enhancement, which could help improve the overall health of the ecosystem. However, we also found that simply increasing the number of a few species is not enough to fully optimize the ecosystem. To improve the food web and the ecosystem as a whole, more comprehensive strategies are needed, such as enhancing protection and management practices, improving artificial reef design and deployment, and boosting ecological connectivity.

## 1. Introduction

Inshore water areas within an Exclusive Economic Zone (EEZ) extend up to 200 nautical miles from the coastal baseline and include both the water column and seabed. These areas are crucial for providing essential ecological services, such as food and raw materials, environmental purification, climate regulation, and cultural enrichment [1,2]. Although they cover only 18% of the Earth’s surface, these areas contribute to 25% of the planet’s primary productivity and account for 90% of global fishery yields [3]. In particular, inshore waters are of vital importance to China, playing a critical role in ensuring food security and maintaining ecological balance [4,5]. Ecological balance is defined as the stable state achieved by ecosystems through development and regulation, characterized by structural stability, functional stability, and a balance in energy input and output [6]. This dynamic equilibrium is crucial as it supports continuous energy flows and material cycles, alongside the renewal of biological entities. However, recent human activities, such as industrialization, urbanization, agricultural expansion, and climate change, have severely degraded these systems. The resulting disruption to China’s inshore habitats and fishery resources has led to reduced fishery yields, diminished biodiversity, and the simplification of food webs [7]. These changes significantly undermine the ecological balance that is essential for sustaining both the productivity and biodiversity of these critical areas.

In response to the challenges posed by the degradation of inshore ecosystems, China has proposed the establishment of a marine ecological civilization and implemented a range of measures aimed at protecting and restoring marine ecosystems. These initiatives include the restoration of nearshore environments, the development of marine ranching projects, the enforcement of seasonal fishing moratoriums, and the regulation of fishing vessels—efforts designed to enhance the stability, diversity, and sustainability of marine ecosystems [8,9]. Among these, the development of marine ranching projects is viewed as a key strategy for transforming and upgrading China’s marine fisheries while safeguarding the marine environment [10].

In China, marine ranching is defined as a fishery model in a specific sea area aimed at rebuilding fishery stocks through measures such as releasing cultured juveniles and restoring habitats while aligning with natural ecosystem processes [11]. It differs from traditional aquaculture and hatchery release. Traditional aquaculture focuses on increasing output through external inputs, emphasizing production and economic benefits, whereas marine ranching emphasizes ecosystem restoration, fully utilizing natural productivity, and prioritizing ecological protection. Compared to pure stock enhancement through hatchery release, marine ranching places greater emphasis on habitat restoration and resource management, which is more conducive to increasing survival and recapture rates of the targeted species [10,11].

The similarities between marine ranching in China and other countries lie in their shared efforts to enhance marine fishery resources and restore ecological environments. The primary methods used include artificial reefs and hatchery release, with a focus on boosting fishery resources and protecting the environment. For example, Japan, South Korea, and the United States enhance marine fishery resources through the deployment of artificial reefs, construction of seaweed beds, and hatchery release; Australia focuses on restoring seagrass beds and coral reefs to conserve marine biodiversity [12]. The differences, however, primarily lie in the industrial models. In Japan and South Korea, marine ranching integrates hatchery release and artificial reef deployment with the establishment of fisheries management centers to enhance fishery resource output. In North America, artificial reef deployment is followed by the development of a large recreational tourism industry, including activities such as diving tourism. European marine ranching, represented by countries like Norway, Germany, and the UK, focuses on hatchery release to increase fishery resources, thereby supporting commercial harvesting [13,14,15,16]. In contrast, China is dedicated to developing a comprehensive industrial framework for marine ranching that integrates site selection, planning, habitat restoration, biomass conservation, and safety assurance, with the aim of enhancing marine fisheries and safeguarding the marine environment. Considerable emphasis has also been placed on the integration of primary, secondary, and tertiary sectors, particularly through the convergence of fisheries and tourism, as well as the alignment of energy and fisheries management [12,17].

Effective assessment of the structural and functional characteristics, as well as the carrying capacity of marine ranching ecosystems, is a fundamental prerequisite for the successful construction and management of marine ranching in China [7]. This is not only crucial for holistic planning but also for ensuring the scientific construction and management of these ecosystems [11,18,19]. Specifically, these assessments provide essential data to guide decisions on the construction area, type, and scale of marine ranching projects. For example, they help in selecting appropriate hatchery release species, estimating optimal stocking densities, and determining the appropriate scale for artificial reef construction [20,21,22,23,24]. Moreover, these assessments are critical for ensuring the long-term sustainability of marine ranching through effective management, which includes determining strategies for sustainable harvest based on carrying capacity [25,26,27].

Because of the limitations in carrying-capacity assessment technology, most marine ranching projects in China have not conducted comprehensive evaluations of carrying capacity [7]. Existing assessments have largely focused on community structure, water quality, and the enhancement effects of individual species [23,28,29,30,31,32,33], with little attention to the overall performance of marine ranching systems [11,34]. Even system-level evaluations generally compare marine ranching with small adjacent areas, lacking a robust evaluation system and clear grading standards [33,35,36,37,38,39], which undermines the credibility of the results. As a result, most marine ranching projects rely heavily on empirical knowledge, lacking a solid scientific foundation for selecting hatchery release species, estimating stocking densities, and designing habitat construction strategies. This reliance leads to significant ecological and economic risks, including low survival rates of stocked species, declines in wild population biomass, depletion of fishery resources, artificial reef subsidence, and hypoxia, as well as wasted financial resources and economic losses due to the high mortality of stocked species [7,10,11,23,40,41,42,43]. Therefore, there is an urgent need for the development of systematic evaluation and simulation methods for marine ranching ecosystems to enhance the scientific rigor and sustainability of marine ranching projects in China.

Ecological Network Analysis (ENA) is a systems-based methodology that quantifies the structure and function of marine ecosystems by analyzing the material and energy flow relationships among all components within ecosystems’ food web [44,45,46,47,48,49,50,51,52]. This approach has proven to be a valuable tool for assessing ecosystem health and for linking ecological and socio-economic systems. It is widely applied in marine ecosystem management and ecological restoration, providing critical insights for decision making and policy development [53,54,55,56].

The Ecopath with Ecosim (EwE) model is widely used to calculate ENA indices for marine ecosystems. It consists of the following three main components: Ecopath, Ecosim, and Ecospace. Ecopath is primarily employed to analyze the material and energy flows among various ecosystem components, while Ecosim simulates the dynamics of ecosystem food webs under different pressures. The EwE model has been extensively applied in recent studies to assess and simulate marine ranching ecosystems [33,35,37,39,57].

In this study, a marine ranching ecosystem and its control counterpart in the Beibu Gulf of China were selected as research subjects. Ecopath models were constructed for both ecosystems to evaluate their energy flow, trophic structure, and ecological carrying capacity. An index system based on ENA and a fuzzy comprehensive evaluation model was developed to assess the status of these ecosystems. Suitable stock enhancement groups for the marine ranching ecosystem were identified, and ecosystem dynamics were simulated under different stock enhancement strategies. This research aims to provide essential technical and theoretical support for the comprehensive evaluation and ecosystem-based stock enhancement of marine ranching. Furthermore, it seeks to contribute to the sustainable and high-quality development of marine ranching in the Beibu Gulf and across China.

## 2. Materials and Methods

### 2.1. Introduction of the Marine Ranching

The marine ranch, named “Jinggong”, is located to the southwest of the Guantou Ridge in Beibu Gulf, with coordinates ranging from 21°25′31.26″ N to 21°25′58.96″ N and from 108°54′15.39″ E to 108°56′38.37″ E. The specific location of the marine ranch is illustrated in Figure 1. Covering an area of 4.80 km^2^, the marine ranch has an average water depth of approximately 9 m. According to the classification standards of the Ministry of Agriculture and Rural Affairs for national marine ranches, the Jinggong Marine Ranch is categorized as a “recreational” type. It was primarily established through artificial reef construction and hatchery releases. Moreover, an online environmental monitoring system was implemented to track the water quality. Since December 2020, approximately 20,000 cubic meters of artificial reefs were deployed, categorized into four types, as shown in Figure 2, with the materials predominantly being composed of reinforced concrete. The primary objective of the marine ranching project is to enhance the conservation and augmentation of fishery resources, thereby ensuring the sustainable development of fisheries. The target species for conservation and enhancement include fish such as Yellowfin Seabream (*Acanthopagrus latus*) and False kelpfish (*Sebastiscus marmoratus*), crustaceans like the Japanese Prawn (*Marsupenaeus japonicus*) and Japanese Stone Crab (*Charybdis japonica*), and Sea cucumber (*Stichopus variegatus*), as well as mollusks such as the Pacific Oyster (*Crassostrea gigas*) and Small Giant Clam (*Lutraria sieboldii*). Additionally, the project aims to stimulate the economy through recreational tourism activities, ultimately promoting the integrated development of both economic and social benefits. The control ecosystem is located about 2 km south of the marine ranch and has a water depth of about 9 m and a sandy silt bottom substrate. The selection of the control site was based on its ecological similarity to the marine ranching, particularly in terms of proximity to land and environmental characteristics. Both areas are located on the western side of Guantou Ridge and exhibit similar water quality conditions. While the ecological features of these areas are comparable, it is recognized that there are still some differences in factors such as flow fields and human activity impacts, particularly due to the ranching area’s closer proximity to the northern mainland.

### 2.2. Construction of Ecopath Models

#### 2.2.1. Introduction of the Ecopath Model

Please refer to the Appendix A: Introduction of Ecopath model.

#### 2.2.2. Survey of the Marine Environment

Surveys of the biotic and abiotic environments in the Jinggong marine ranching and control ecosystems were conducted in spring and autumn of 2023. Phytoplankton biomass was assessed by measuring chlorophyll a according to standard procedures [58]. Zooplankton samples were obtained through vertical tows using plankton nets with a mesh size of 169 µm.

Benthic and swimming organisms in mud substrate areas were surveyed following the Specifications for Marine Surveys GB/T12763.6-2007 [59]. Meanwhile, the biomass of swimming animals (e.g., fish and cephalopods) was calculated using the swept-area method [60]. The hauling speed was approximately 2.5 knots, with an inner mesh size of 3.5 cm and a mouth width of 6 m. Each trawl lasted for 30 min. There were 3 survey sites in the marine ranch and 9 survey sites in the control area.

Swimming organisms and macrobenthos in reef areas were surveyed using SCUBA diving techniques (GoPro, San Mateo, CA, USA) for video recording and hand collection [37,61]. Meanwhile, the survey of fish in the reef area was conducted using a transect method combined with cage net sampling. After the diver descended into the water, they performed transect-based swimming video recording. The video duration at each station was 15 min, with an average sweeping area of 18 square meters per station. Biomass was calculated as the number of each fish species observed multiplied by the average weight derived from the cage net, divided by the swept area of the video recording. Those same transects were used for mobile invertebrates (mainly crustaceans and cephalopods). Macrobenthos such as sea urchins and gastropods were collected from reefs using 0.5 × 0.5 m quadrats, while attached organisms, such as mussels, barnacles, and oysters, were collected by scraping with a small knife. Six stations were set in the reef area. Trawl surveys were also conducted around the reef area; this was primarily aimed at obtaining the biomass of species such as small shrimp, which are difficult to detect and capture with video and cage nets. The taxa, biomass, and abundance of zooplankton, benthic, and swimming animals were measured for all collected samples.

#### 2.2.3. Functional Group Division

Adhering to the functional group division principle of the Ecopath model [62], the marine ranching ecosystem was categorized into 23 functional groups, including pelagic fishes, large and medium demersal fishes, sparids, leiognathidae, small demersal fishes, scorpaenidae, gobiidae, mantis shrimps, large crabs, other crabs, *Metapenaeopsis barbata*, other shrimps, cephalopods, sea urchins, gastropods, barnacles, oysters, mussels, other bivalves, other benthos, zooplankton, phytoplankton, and detritus. The phytoplankton group served as the primary-producer group, while all other groups functioned as consumers, except for detritus. In contrast, the control ecosystem comprised 19 functional groups, which exhibit similar compositions to the marine ranch but excluding sparids, barnacles, oysters, and mussels. The specific species of each functional group are listed in Appendix E: Tables (Table A1 and Table A2).

#### 2.2.4. Data Sources of Functional Groups

The Ecopath models (Ecopath with Ecosim 6.6.7) were designed to model a period of 1 year. The biomass for each functional group is expressed as the wet weight in t/km^2^. The input data for Biomass (B), Production-to-Biomass ratios (P/B), Consumption-to-Biomass ratios (Q/B), and diet composition of each group were estimated using data obtained from field surveys and literature sources. The methods employed to acquire these data are detailed in the Appendix E: Tables (Table A3). The diet composition of each consumer group was displayed in Table A4 and Table A5. The Unassimilated Ratio of Consumption (Ui) for mussels, barnacles, oysters, and other bivalves was set at 0.40, while the group of other benthos was set at 0.35, and all other consumer groups were set at 0.20 [62,63,64]. Fishery data were provided by the Qinzhou Agriculture and Rural Bureau of Guangxi Zhuang Autonomous Region.

#### 2.2.5. Model Balancing and Uncertainty

We used the estimated Ecotrophic Efficiency (EE) value of each functional group (which should be <1) as the primary criterion for model calibration. If the estimated EE exceeded 1, indicating that the consumed biomass surpassed the produced biomass, we incrementally adjusted the diet composition of each consumer group, with each adjustment not exceeding 0.05, to reduce the EE value below 1. Furthermore, we ensured that most of the P/Q values (the gross food conversion efficiency: ratio between production and consumption) were in the range of 0.1–0.3. We also ensured that the Respiration-to-Assimilation (R/A) and Production-to-Respiration (P/R) ratios in the model were <1; the Respiration-to-Biomass (R/B) ratio was higher in active species than in sedentary groups [62,65,66]. The pre-balanced diagnosis was also used to identify issues in the model’s structure and in the data quality prior to balancing the models [65].

### 2.3. Construction of Indices System

#### 2.3.1. Description of ENA Indices

This study established an index system to evaluate the ecosystem status of marine ecosystems based on ENA indices. The ENA indices were categorized into three groups. Firstly, indices representing the function of the ecosystem included Detritivory/Herbivory (D/H), the average Transfer Efficiency among different trophic levels (TE), and relative Ascendancy (A/C). Secondly, indices representing the characteristics of the ecosystem food web’s structure include the Connectance Index (CI), System Omnivory Index (SOI), Finn’s Cycling Index (FCI), and Average Path Length (APL) [67]. Lastly, indices representing the maturity of the ecosystem include Total Primary Production/Total Respiration (TPP/TR), Total Primary Production/Total Biomass (TPP/TB), and Total Biomass/Total System Throughput (TB/TST) (Table 1). The meanings of each index are described in Appendix C: Introduction of Ecological Network analysis indicators.

#### 2.3.2. Classification of Ecosystem Status Levels

Due to the lack of historical data on ENA indices in the study areas, directly evaluating the current status of the ecosystems was challenging. An alternative approach was to establish standards through inter-ecosystem comparison. Therefore, published literature on ecosystem assessments based on ENAs for the world’s coastal marine ecosystems (139 ecosystems in total, including estuaries, bays, islands, straits, oyster reefs, and artificial reefs) was collected, as much as possible, and is presented in Appendix E: Tables (Table A6). The ecosystems were divided into 5 levels using the quintiles method. For positive-type indicators, where a higher value indicates a better system performance, the first, second, third, fourth, and fifth quintiles denote the critical values for the “poor”, “relatively poor”, “medium”, “relatively good”, and “good” grades, respectively. For negative-type indicators, where a higher value indicates a worse system performance, the first, second, third, fourth, and fifth quintiles denote the critical values for the “good”, “relatively good”, “medium”, “relatively poor”, and “poor” grades, respectively (Table 2).

Among the indices, D/H, TE, TB/TST, CI, SOI, FCI, and APL indicate a better ecosystem status as their values increase. Regarding A/C, Ulanowicz et al. (2009) proposed that the optimal trade-off value is 0.4596 [68]; values below this threshold are positively correlated with an improved system performance. Since none of the cases examined in this study exceeded this threshold, A/C was considered a positive indicator. For TPP/TR and TPP/TB, values greater than 1 suggest a lower ecosystem maturity. The weighting of the indicators follows the methodology outlined by Zeng et al. (2021) [69]. As some of the Ecosim scenarios in our study did not include fishing activities, we excluded the mean TL of the catch indicator, as considered by Zeng et al. (2021) [69], and proportionally redistributed its weight among the remaining ecosystem function indices.

#### 2.3.3. Evaluation of the Ecosystem Status of the Ecosystems

Fuzzy Comprehensive Evaluation (FCE) was used to calculate the composite score of all indicators. The final evaluation result was determined based on the principle of maximum membership degree. The specific evaluation steps are referenced from the methods of Tobor-Kapłon et al. (2007), Dong et al. (2021), and Wu and Hu (2020) [70,71,72].

### 2.4. Evaluation of Ecological Carrying Capacity

Ecological carrying capacity is defined as the maximum biomass that functional groups within an ecosystem can sustain while maintaining an energy balance [73]. Specifically, energy balance refers to a state in which the energy inputs and outputs within an ecosystem are in equilibrium [74]. The method for evaluating the ecological carrying capacity was adapted from Jiang and Gibbs (2005) [75]. The biomass of the target functional group was incrementally increased until the ecosystem became unbalanced. The critical point immediately before ecosystem disbalance was identified as the ecological carrying capacity. No parameters other than the biomass of the target functional group were manually altered during the calculation of the ecological carrying capacity.

### 2.5. Evaluation of Stock Enhancement Potential and Selection of Stock Enhancement Groups

All functional groups in the marine ranch were categorized into the following three groups based on their Trophic Levels (TLs): TL 2.0–2.5, TL 2.5–3.0, and TL 3.0–3.5. The difference between the ecological carrying capacity and the current biomass of each functional group was used to assess its stock enhancement potential, with a larger difference indicating higher potential. Within each TL category, groups with higher stock enhancement potential were identified as candidates for stock enhancement. However, the maturity of relevant enhancement technologies, such as seedling breeding and larval releasing, was also considered critical for the final selection. Groups with high enhancement potential but underdeveloped enhancement technologies were excluded from the stock enhancement groups, and only those groups with both high enhancement potential and mature enhancement technologies were selected.

### 2.6. Simulation of Stock Enhancement Strategies

The Ecosim model was employed to simulate the dynamics of the ecosystem food web over the next 14 years under different stock enhancement strategies. In each simulation scenario, the Ecopath model from the tenth year was extracted to represent the new state of the ecosystem. The ecosystem status was then evaluated using the indices system based on the Ecopath model’s ENA indices.

#### 2.6.1. Introduction of the Ecosim Model

Please refer to Appendix B: Introduction of the Ecosim Model.

#### 2.6.2. Construction of Ecosim Model

The Vulnerability (*v*)-index is a critical parameter in constructing the Ecosim model. Which determines whether the trophic control between predator and prey is a top-down, bottom-up, or intermediate effect. Because of the lack of historical survey data in the marine ranch, the following empirical formula was applied to calculate the *v* index for each functional group [76,77]:(1)$$v_{i}=0.1515\times TL_{i}+0.0485$$
where *TL_i_* is the *TL* corresponding to functional group *i*. The *v* settings ranged from 0 to 1, with 0.0–0.3 representing bottom-up control, 0.3 representing mixed control, and 0.3–1.0 describing a top-down impact [78]. The *v_i_* was then transformed to derive *v_new_* for Ecosim input, which ranged from 1 to ∞, as follows:(2)$$\log\left(v_{new}\right)=2.301958\times v_{i}+0.001051$$

Within the marine ranching ecosystem, oysters, barnacles, and mussels act as ecosystem engineers, enhancing the spatial complexity and heterogeneity of the habitat. They provide essential refuges, foraging, and reproductive spaces for organisms in the artificial reef area, which in turn influence the behavior and distribution of predators. These effects were incorporated into the simulations using the mediation functions in Ecosim. Sigmoidal functions were used to modulate the predator–prey relationships through these mediation functions in this study. The specific mediation settings were determined based on methodologies outlined by Harvey (2014) and Sadchatheeswaran et al. (2020) [79,80].

#### 2.6.3. Simulation Scenario Design

We assumed that the maximum biomass achievable by the selected stock enhancement group within the marine ranching ecosystem corresponds to the biomass at the ecological carrying capacity. Therefore, setting the biomass of the stock enhancement group at the ecological carrying capacity became one of the driving factors in establishing the Ecosim model. Additionally, fishing effort was also taken into account as a driving factor for the model’s construction. The fishing data used to drive the model were obtained from the fishing efforts in the marine ranch during 2023.

Based on the selection of the stock enhancement groups, the following three species were chosen: mussels, large crabs, and scorpaenids. The method for enhancing mussels in the marine ranch involved constructing artificial reefs, which would provide them with a suitable habitat. However, once the reefs were deployed, not only mussels but also oysters and barnacles grew on them, as these three groups share similar ecological characteristics. As a result, the groups being enhanced were these three. To better simulate the effect of this enhancement method, we included all three of these groups in the enhancement simulation and combined mussels, oysters, and barnacles (MOB) into a single stock-enhancement group.

Three single-group stock enhancement strategies were established for MOB, large crabs, and scorpaenidae. Additionally, the following four multiple-group stock enhancement strategies were set: MOB + large crabs, MOB + scorpaenidae, large crabs + scorpaenidae, and MOB + large crabs + scorpaenidae. For each stock enhancement strategy, the following two scenarios were simulated: with fishing and without fishing.

For with fishing activity, a constant fishing effort was applied throughout the simulation, remaining unchanged over time. The fishing effort was set to match the catches from the marine ranch in 2023. Given that the marine ranching ecosystem had reached a relatively stable state by 2023, this catch level is considered a reasonable reflection of the fishing pressure during the stock enhancement process.

For without fishing activity, the fishing effort was completely excluded in this scenario, and stock enhancement was simulated without any fishing pressure. This allowed for the evaluation of the effects of stock enhancement in the absence of fishing activity.

An additional simulation scenario was also included, involving only fishing activity, where fishing effort was applied without any stock enhancement. Therefore, a total of 15 simulation scenarios were conducted in this study.

## 3. Results

### 3.1. Trophic Structure

The TLs of the functional groups in the marine ranching ecosystem ranged from 1 to 3.46 (Table 3 and Table 4). Sparids exhibited the highest TL, followed by large- and medium-sized demersal fishes and scorpaenidae (3.42 and 3.27, respectively). Additionally, cephalopods, small-sized demersal fishes, and mantis shrimps also displayed relatively high TLs (3.28, 3.22, and 3.22, respectively). Conversely, oysters and mussels had low TLs, both at 2.02. In the control ecosystem, the functional groups exhibited TLs ranging from 1 to 3.63, with sparids being the highest, followed by large- and medium-sized demersal fishes and cephalopods (3.35 and 3.25, respectively). Scorpaenidae and small-sized demersal fishes also had relatively high TLs (3.30 and 3.16, respectively).

The TL I comprised phytoplankton and detritus, TL II mainly consisted of shellfish, barnacles, shrimps, zooplankton, and sea urchins, while TL III mainly consisted of fish, cephalopods, mantis shrimps, and crabs in both ecosystems. Mussels had the highest biomass in TL II (80.40 t/km^2^), followed by barnacles (30 t/km^2^), while sparids had the highest biomass in TL III (0.33 t/km^2^), followed by large crabs and other crabs (both at 0.30 t/km^2^) in the marine ranching ecosystem. The total biomasses distributed among TLs I, II, and III in the marine ranching ecosystem were 60.29, 153.17, and 1.92 t/km^2^, respectively, compared to 65.67, 7.00, and 1.26 t/km^2^, respectively, in the control ecosystem. The phytoplankton showed the highest biomass in the control ecosystem (24.64 t/km^2^).

### 3.2. Energy Flow Structure

The EE values of the functional groups in the two ecosystems are presented in Table 3 and Table 4. In the marine ranching ecosystem, large crabs, mantis shrimps, and cephalopods exhibited high EE values of 0.92, 0.86, and 0.80, respectively. Conversely, sea urchins, barnacles, oysters, and mussels displayed very low EE values due to their high biomass and lack of predators. Shrimps exhibited the highest EE value (0.96) in the control ecosystem, followed by other bivalves, other crabs, *M. barbata*, and other benthic organisms, with EE values of 0.95, 0.92, 0.91, and 0.89, respectively. Leiognathidae, zooplankton, and detritus had very low EE values (0.01, 0.07, and 0.09, respectively).

The energy flow among the TLs in the two ecosystems is depicted in Figure 3. Approximately 2152.00 and 2365.00 t/km^2^/a of energy flowed to TL II in the marine ranching and control ecosystems, respectively, accounting for 68.00% and 16.00% of their total primary production, respectively. The transfer efficiencies between TL II and III and III and IV were 4.00% and 6.00% in the marine ranching ecosystem, respectively, they were 4.00% and 11.00%, respectively. The average transfer efficiencies among TLs II to V were 5.84% and 6.47% in the marine ranching and control ecosystems, respectively.

### 3.3. Ecosystem Attributes

The metrics of the Total System Throughput (TST), total production, and total biomass—key indicators of ecological size—were 2.75, 1.40, and 5.56 times higher, respectively, in the marine ranching ecosystem compared to the control ecosystem (Table 5). In the marine ranching ecosystem, the proportion of total consumption to TST was the highest at 39.06%, followed by the flow to detritus, which accounted for 30.56% of the TST. In contrast, in the control ecosystem, the highest proportion was the flow to detritus at 72.59% of the TST, followed by total consumption, which accounted for 19.08%.

### 3.4. Ecosystem Status

The values of the ENA indices of the marine ranching and control ecosystems are presented in Table 6. The evaluated results of the ecosystem statuses of the marine ranching and control ecosystems are presented in Table 7. The ecosystem status of the marine ranching ecosystem was rated as “relatively good”. Specifically, the FCI was classified as “good”, while the FML, TPP/TR, TPP/TB, and TB/TST were classified as “relatively good”. The CI and D/H were classified as “medium”, and the SOI, A/C, and TE were classified as “relatively poor”. In contrast, the ecosystem status of the control ecosystem was rated as “relatively poor”. Specifically, the CI and SOI were classified as “relatively good”, while D/H was classified as “medium”. Furthermore, the FCI, FML, A/C, TPP/TR, and TPP/TB were all classified as “relatively poor”.

### 3.5. Ecological Carrying Capacity and Stock Enhancement Potential

The ecological carrying capacities of the two ecosystems are detailed in Table 8. Notably, the functional groups within the marine ranching system exhibited generally higher carrying capacities compared to those in the control ecosystem. Mussels displayed the highest carrying capacity within the marine ranch at 163 t/km^2^, closely followed by oysters at 96 t/km^2^. Pelagic fishes emerged as the group with the highest carrying capacity among fish species. Conversely, the control ecosystem’s highest carrying capacity was observed with shrimps at 0.95 t/km^2^, followed by large- and medium-sized demersal fishes at 0.48 t/km^2^.

The stock enhancement potential of the economic functional groups within the marine ranching system was estimated based on the disparity between the carrying capacity and current biomass (Table 9). Among groups with TLs of 3.0–3.5, gobiidae exhibited the highest stock enhancement potential, followed by scorpaenidae. In the TL range of 2.5–3.0, pelagic fishes demonstrated the greatest potential, followed by large crabs. For TLs between 2.0 and 2.5, mussels showcased the highest potential, followed by oysters.

Currently, there are no effective stock enhancement technologies available for gobiidae and pelagic fishes. However, seedling breeding and larval releasing technologies for large crabs (mainly composed of *Portunus trituberculatus*) and scorpaenidae (mainly composed of *Epinephelus moara*) have reached a mature stage [81,82], this study opted for large crabs and scorpaenidae as suitable candidates for stock enhancement. Moreover, the enhancement of mussels is primarily hindered by the limited availability of hard substrate within the marine ranch. Therefore, constructing artificial reefs can effectively enhance mussel populations. Recognizing that mussels, oysters, and barnacles (MOB) thrive upon the deployment of artificial reefs, these three groups were amalgamated into a single-stock enhancement group, with an estimated carrying capacity of 218.06 t/km^2^.

### 3.6. Ecosystem Dynamics Under Different Stock Enhancement Strategies

The Ecosim model was employed to simulate the dynamics of the marine ranching ecosystem under various stock enhancement scenarios for the forthcoming 14 years. The outcomes are illustrated in Appendix F: Figures (Figure A1, Figure A2, Figure A3, Figure A4, Figure A5, Figure A6, Figure A7, Figure A8, Figure A9, Figure A10, Figure A11, Figure A12, Figure A13, Figure A14 and Figure A15). Table 10 and Table 11 present the ecosystem status of the marine ranching system at the tenth year of modeling. The results reveal that in scenarios involving fishing alone, stock enhancement of MOB + fishing, scorpaenidae + fishing, and MOB + scorpaenidae + fishing, large crabs experienced a rapid decline in biomass to zero. Similarly, in scenarios of large crabs + fishing and large crabs + scorpaenidae + fishing, the biomass of large and medium demersal fishes also decreased to zero. These observations suggest a collapse in the energy flow structure within these scenarios. Consequently, only the ecosystem statuses of predicted scenarios without such collapse can be effectively evaluated using the index system. The evaluated results indicate that in scenarios involving the stock enhancement of MOB, MOB + large crabs, MOB + large crabs + fishing, MOB + scorpaenidae, MOB + large crabs + scorpaenidae, and MOB + large crabs + scorpaenidae + fishing, the ecosystem status notably improved to a “good” level. Specifically, there was an observed decrease in the indices of TPP/TR and TPP/TB, accompanied by observed increases in the FCI and FML, while other indices displayed minimal changes.

## 4. Discussion

In China, the primary objective behind developing marine ranching was to restore the marine environment and ensure a sustainable yield of fishery resources [83]. Following almost four years of development, the Jinggong marine ranch has experienced significant transformations in both biotic community composition and system functionality compared to the control ecosystem. The biomasses of fish, crustaceans, and mollusks within the marine ranch are 1.82, 16.99, and 33.51 times greater, respectively, than those observed in the control ecosystem.

Artificial reefs play a crucial role in significantly increasing the mollusk biomass by providing essential substrates for attached organisms such as mussels and oysters [84,85,86]. These mussels and oysters, through their robust filter-feeding and biodeposition activities, transfer substantial amounts of particulate organic matter to benthic environments, thereby enriching these habitats with essential nutrients [87]. Furthermore, artificial reefs modify the flow field, creating slowed currents within the reef structure [88,89]. This alteration increases the deposition rate of particulate organic matter and further enhances the nutrient content in the reef’s benthic environment. Consequently, the elevated nutrients stimulate the activity of benthic microorganisms, accelerating the transformation and cycling of nutrients at the water’s bottom [90]. These processes provide additional nutrients for benthic microalgae and enhances their primary productivity [91,92]. Additionally, artificial reefs can generate upwelling, which transports nutrients from the bottom to upper water layers, thus accelerating nutrient cycling and enhancing the primary productivity of pelagic phytoplankton [93,94]. Moreover, the significant spatial heterogeneity provided by artificial shellfish reefs offers effective refuges and foraging grounds for marine organisms [93,95,96]. These functions of artificial reefs—particularly accelerating nutrient cycling, enhancing primary productivity, and providing habitats for protection, foraging, and breeding—are crucial in supporting the growth and reproductive success of various marine organisms. Ultimately, these activities significantly enhance the biodiversity and biomass of marine resources [97,98].

The SOI and CI serve as crucial indicators of ecosystem food web complexity. In the marine ranching and control ecosystems, the CI values were 0.28 and 0.32, respectively, while the SOI values were 0.13 and 0.21, respectively. Comparatively, the CI in the marine ranching system was rated at a “medium” level in relation to those of the coastal marine ecosystems collected in this study, while the SOI was rated as “relatively poor”. Moreover, both the CI and SOI values were notably lower compared to the small marine ecosystems (1–10 km^2^) investigated by Heymans et al. (2014) [51]. The CI reflects the ratio of actual links to potential links within an ecosystem’s food web, whereas the SOI delineates the distribution of feeding interactions across TLs [49,62]. The SOI compensates for the limitations of the CI in delineating food web complexity, especially given that the CI remains constant despite variations in the prey proportions within predator diets [62,99]. The low SOI observed in the marine ranching ecosystem suggests weak connections among functional groups, potentially resulting in lower energy flux and a simplified food web structure. The lower CI and SOI in the marine ranch compared to the control ecosystem may stem from increased biomasses of mollusk, echinoderm, and barnacles. The simpler diets of these species contribute to a less complex food web structure. The increased biomass of these groups also contributed to increased EE values for phytoplankton and detritus within the marine ranching ecosystem, resulting in a notable enhancement of energy utilization efficiency between TLs I and II. However, despite these improvements, the TE values among TLs II to V remained notably low, averaging 5.84%. This level was classified as “relatively poor” according to the index system, falling far below the natural ecosystem average of 10% [100]. Optimizing the food web to facilitate smoother energy transfer pathways and increasing the biomass of the higher TL organisms (such as those at TLs III and IV) are two essential aspects of improving TE values across TLs II to V. Mid-TL organisms, including crustaceans, cephalopods, and bivalves, play a pivotal role as intermediaries in the energy transfer between primary producers and apex predators [101,102,103]. Employing strategies such as habitat restoration, biological conservation, and stock enhancement to increase the diversity and biomass of mid-TL organisms may help optimize the food web and increase the biomass of higher TL organisms, thereby improving the TE values across TLs II to V.

The A/C serves as an indicator of ecosystem organization and efficiency. Both ecosystems in this study were classified at the “relatively poor” level, suggesting diminished organization and system efficiency. The observed low A/C value in the marine ranching ecosystem may be attributed to the fact that increasing system organization levels hinge on the succession of community structure, a process that typically unfolds over an extended period [104,105]. Additionally, external disturbances, such as fishing activities and marine natural disasters, can disrupt the self-organization process of the ecosystem [106]. For instance, massive waves and strong winds caused by typhoons can disturb seabed sediments, leading to a significant decrease in the abundance and species composition of large benthic organisms within a short period [107]. This, in turn, hinders the ecosystem’s ability to achieve higher levels of organization.

The low complexity of the food web, TE, and A/C values in marine ranching suggest that this system may be subject to significant external pressures [51,108,109,110], resulting in the majority of the TST occurring at low TLs. Overfishing emerges as a prominent source of pressure among various external stressors. Indeed, overfishing has led to a severe decline and, in some cases, the depletion of large carnivorous fish in China’s coastal ecosystems [7,111], echoing the phenomenon of “fishing down the marine food web” observed in diverse contexts [112,113,114]. Modeled results from the Ecosim in this study indicate that large crabs in the marine ranch are subject to overfishing. Chen et al. (2008) similarly observed overfishing in the Beibu Gulf [115]. Fishing activities in areas adjacent to marine ranching may adversely affect marine ranching. The migration patterns of many marine creatures within and around these ranches, particularly for types II and III fish species that only spend part of their lifecycle in the ranches, expose them to significant fishing risks during their movement among regions [11,116,117,118]. Considering the diverse habitat requirements of marine creatures at different life stages, proposals such as those by Yang and Ding (2022) are noteworthy [119]. They suggest constructing global aquatic ecological ranching systems that designate entire estuaries or bays as ranching areas. Similarly, Liang et al. (2020) propose developing marine ranching facilities designed to meet all habitat needs across different life stages of fish [118]. In the long term, these proposals hold the potential to mitigate challenges faced by marine ranching, including strong external disturbances and difficulties in maintaining a complex food web structure and high system organization levels.

The TPP/TR and TPP/TB ratios in the marine ranching and control ecosystems were 1.82 and 8.93, and 17.27 and 85.22, respectively. These values in the marine ranch were rated as “good”, whereas those in the control ecosystem were deemed “relatively poor”. Additionally, the D/H ratio in the marine ranch exceeded that in the control ecosystem. These findings suggest that the establishment of marine ranching has significantly enhanced ecosystem maturity and stability [120,121]. Despite comparable total primary production in both ecosystems, the decline in the TPP/TR and TPP/TB ratios can be attributed to increases in the total biomass and total respiration. Notably, in marine ranching, filter-feeding bivalves accounted for the largest portion of both the total biomass and total respiration. These bivalves exhibit ecological characteristics similar to those of zooplankton but with lower turnover rates. While zooplankton can quickly respond to ecosystem disturbances, bivalves exhibit slower responses, thus creating a pathway for slow energy flow. The asynchronous dynamics of bivalves and zooplankton likely play a pivotal role in sustaining ecosystem stability [122].

The evaluation based on the index system revealed that the ecosystem status in marine ranching was “relatively good”, contrasting with the “relatively poor” status of the control ecosystem. Notably, marine ranching’s maturity indices, such as TPP/TR, TPP/TB, and TB/TST, significantly outperformed those of the control ecosystem. Furthermore, indices like the FCI and FML demonstrated marked improvements, indicating that marine ranching enhances system maturity, energy recycling efficiency, and food chain length. Moreover, metrics such as TST, total production, and total biomass in marine ranching were 2.75, 1.40, and 5.56 times higher, respectively, than those in the control ecosystems. The carrying capacities for fish, crustaceans, and shellfish have also significantly increased in marine ranching areas, these may indicate the positive impacts of marine ranching construction efforts. However, despite these achievements, the construction efforts have yet to effectively improve metrics such as the TE, level of system organization, and food web complexity. Although the results indicate positive effects of marine ranching, it is important to interpret these findings with caution. Differences in factors such as terrestrial inputs and flow fields between the ranching and control areas could influence the outcomes. Future studies incorporating multiple control sites would be valuable for a more robust comparison.

Stock enhancement serves as a pivotal approach to improving marine environments and achieving sustainable utilization of fishery resources [123]. However, the efficacy of stock enhancement in restoring marine resources varies across different initiatives [23,124,125,126,127]. Some studies even contend that this method is entirely ineffective [128]. Simulation scenarios conducted in this study revealed that stocking single fish or crab species did not notably optimize the ecosystem. However, the stock enhancement of the bivalve-dominated MOB group enhanced the ecosystem status from “relatively good” to “good”. This may be attributed to the higher enhancing capacity of MOB compared to crab and fish groups. Additionally, the augmentation in biomasses of mussels, barnacles, and oysters expanded shellfish reefs, providing increased refuge and food resources for crustaceans, cephalopods, fish, and other organisms [86]. This indirect effect also contributed positively to optimizing the ecosystem status.

Yang et al. (2023) demonstrated that stock enhancement of multiple species may have better ecological effects than single-species stock enhancement [129]. This finding aligns with the results of our study. For instance, in scenarios involving single-group stock enhancement of crab, fish, and the MOB group alongside fishing activities, the populations of large crabs or large and medium demersal fishes rapidly collapsed. Conversely, in scenarios combining MOB with crab and fishing, or MOB with crab, fish, and fishing, all functional groups maintained relatively stable biomasses, leading to an increase in ecosystem status to a higher level (from “relatively good” to “good”). These findings suggest that stock enhancement involving multiple species from different TLs represents a more effective strategy for optimizing ecosystem structure and function.

It is important to note that maintaining the biomass of stock enhancement groups at the group’s carrying capacity in simulation scenarios may not fully reflect real-world conditions. The first point is that the estimated ecological carrying capacity was derived based on specific assumptions. The carrying capacity was calculated using a static energy flow model, which represents a theoretical value derived from an ecosystem’s energy balance perspective. This model does not account for the dynamics of biological growth and migration within different functional groups, nor does it consider the impacts of environmental fluctuations [38], Particularly in ecosystems where the biomass or density of a single species increases significantly, issues such as disease outbreaks and interspecies competition may arise [130,131,132,133]. Additionally, the marine ranching area is located near the coast, with a water depth of approximately nine meters. This may support the relatively high primary productivity of benthic microalgae [134]. However, the Ecopath model established did not consider the primary productivity of benthic microalgae, leading to an underestimation of the overall ecosystem’s primary productivity, as well as the carrying capacity of the groups that feed on benthic microalgae. Moreover, the carrying capacity of each functional group is dynamically influenced by external factors, such as climate change and human activities [25]; therefore, it is essential to consider the dynamic nature of carrying capacity in real-world stocking activities [24,38,135]. These factors may lead to discrepancies between the estimated carrying capacity and the actual situation.

The second point is that the instability of the ecosystem increases as it nears its carrying capacity, which makes its dynamic changes difficult to predict. As ecosystems approach this limit, resource constraints, such as prey availability and habitat space, become more pronounced, potentially diminishing their resilience. These increased limitations may heighten the ecosystem’s sensitivity to disturbances, leading to greater population volatility and instability [136,137,138]. Under such conditions, even minor external disturbances or internal fluctuations could push the ecosystem into a state of energy imbalance, significantly elevating ecological risks.

Consequently, these might lead managers to engage excessively in stock enhancement activities, thereby propelling the ecosystem toward a state of energy disequilibrium. This disequilibrium could, in turn, precipitate significant mortality events within the enhanced species and result in the collapse of other functional groups. Therefore, when conducting stock enhancement activities, adopting cautious adaptive management strategies is essential [17]. Under such strategies, integrating the ecological roles of various organisms, implementing multispecies stock enhancements [127], or reducing the stocking density of target organisms may help mitigate these risks.

This study aims to conduct stock enhancement with the objective of maintaining the biomasses of the enhanced groups at their carrying capacity. However, while this strategy maximizes the biomasses of the target groups, factors such as resource availability (e.g., food and space) and disease may prevent the attainment of the Maximum Sustainable Yield (MSY). According to Schaefer (1954), Legović and Perić (1984), and Clark (1990) [139,140,141], MSY is typically achieved when a population reaches half of its carrying capacity. However, this conclusion is based on the assumption of an isolated population growing according to logistic growth models, which overlooks interspecies interactions and food web complexities [142]. As a result, the carrying capacity derived from this approach may exceed the ecological carrying capacity estimated in this study while potentially falling short of the production carrying capacity, which represents the maximum level of production [143,144].

To achieve the MSY for key groups in marine ranching, it may be beneficial to implement appropriate harvesting activities for groups such as sparids and large and medium demersal fishes, whose biomasses are approaching their carrying capacity, thus eliminating the need for stock enhancement. In contrast, species such as large crabs, scorpaenidae, and oysters still maintain a biomass well below half of their carrying capacity, indicating considerable potential for stock enhancement. Therefore, to mitigate the risks associated with such enhancement efforts, it would be prudent to manage the biomass of these groups at approximately half of their ecological carrying capacity.

In scenarios focused on MOB (excluding MOB + fishing and MOB + fish + fishing), the ecosystem status was elevated to a “good” level, indicating that the ecosystem structure and function were effectively optimized compared to the initial state. However, this does not imply that the ecosystem has reached an ideal or perfect state. The index system we constructed in this study provides a general overview of coastal ecosystems, and achieving a ’good’ level does not necessarily align with the standards of mature ecosystems as described by Odum (1969) [120]. For example, the values of TE, A/C, CI, and SOI showed only slight changes in these scenarios, remaining at “relatively poor” or “medium” levels. This indicates that the simulated stock enhancement strategies had a limited impact on optimizing the food web structure, system organization level, and energy transfer efficiency. This finding aligns with the conclusions of Yang et al. (2023) [129], who reported that stock enhancement did not significantly increase ecosystem species diversity, and its restoration effects were limited. It can be inferred that stock enhancement of only a few species may not be sufficient to fully optimize system structure and function. In line with this, Hemraj et al. (2024) emphasized the importance of prioritizing protection and natural recovery to enhance ecosystem structure and function [145]. To truly optimize the structure and function of marine ranching ecosystems, it is crucial to adopt a more integrated approach that includes strengthening ecological connectivity between marine ranching and adjacent areas, improving fishing management strategies, and advancing the design and deployment of artificial reefs [27,146,147,148]. When implemented together, these measures could provide a more sustainable and holistic framework for ecosystem restoration and management in marine ranching.

## 5. Conclusions

In this study, we developed two Ecopath models to assess the energy flow and trophic structure of both marine ranching and control ecosystems. Additionally, we established an index system based on ENA and the fuzzy comprehensive model, incorporating ENA data from 139 coastal marine ecosystems to evaluate the ecosystem status. The results provide valuable insights for the ongoing development of marine ranching initiatives. Importantly, this index system addresses the challenge of lacking reliable standards for evaluating marine ranching ecosystems, offering a methodological framework that can support the assessment of other marine ranching projects in China. The ecosystem status was rated as “relatively good” in the marine ranching ecosystem and “relatively poor” in the control ecosystem. The TST, total production, total biomass, and ecological carrying capacity of economic groups were significantly higher in the marine ranching ecosystem compared to the control, which may indicate the success of the construction efforts. Based on their biomass enhancement potential, mussels, large crabs, and scorpaenidae were identified as the most suitable groups for stock enhancement. Future research should focus on determining the optimal stocking densities by accounting for the mortality rates and growth characteristics of stock enhancement species and exploring the optimal stock levels that would ensure the maximum sustainable yield. The scenario simulations conducted using Ecosim revealed that stock enhancement strategies involving MOB (excluding MOB + fishing and MOB + fish + fishing) elevated the status of marine ranching to a “good” level. Furthermore, enhancing multiple species simultaneously appears to yield better ecological outcomes than focusing solely on single-species stock enhancement. In the simulation scenarios, the biomasses of stock enhancement groups were assumed to reach their carrying capacity levels, which may not fully reflect real-world conditions. Future research should focus on exploring the dynamics of the ecological carrying capacity across different temporal scales and adjusting stock enhancement strategies accordingly to achieve the most effective ecological outcomes. The scenario simulations suggest that stock enhancement involving only a few species is insufficient to fully optimize ecosystem structure and function. To address this limitation, future research should adopt a more integrated approach, focusing on strengthening protection and management strategies, as well as advancing the design and deployment of artificial reefs. These measures could play a pivotal role in enhancing ecosystem structure and function, ultimately ensuring the long-term sustainability and resilience of marine ranching ecosystems.

## Figures and Tables

**Figure 1 animals-15-00165-f001:**
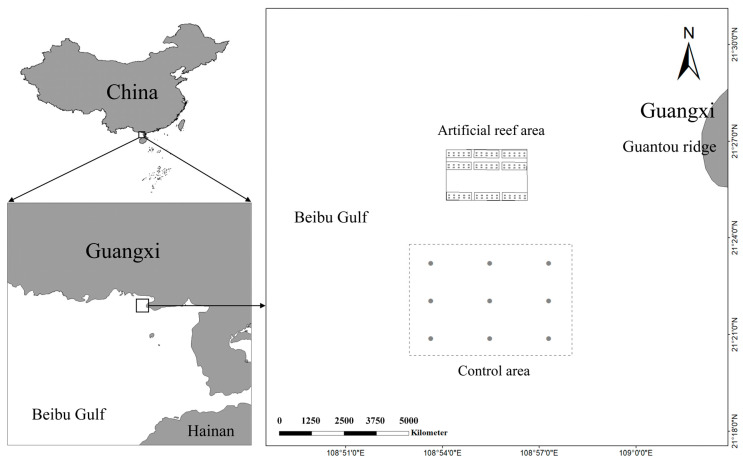
Geographic location and sampling sites of the Jinggong marine ranching and control ecosystems in the Bay of Beibu gulf.

**Figure 2 animals-15-00165-f002:**
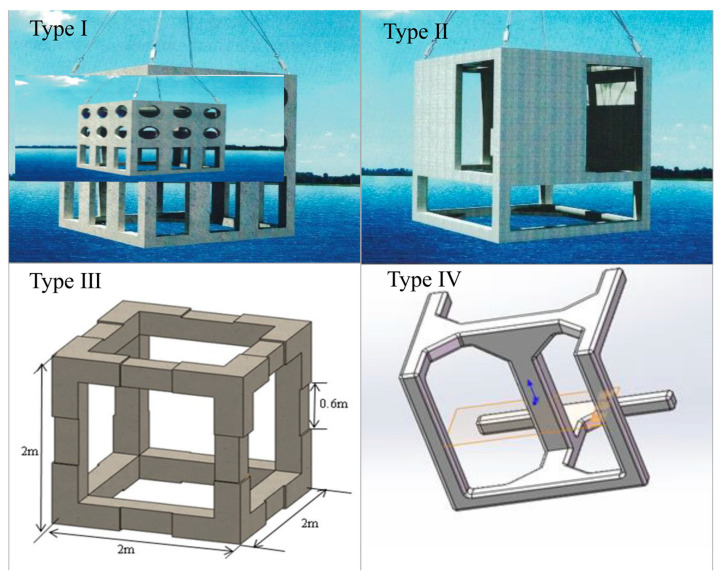
Types of artificial reefs deployed in marine ranching (sizes—type I: 3.0 m × 3.0 m × 3.5 m; type II: 3.0 m × 3.0 m × 3.5 m; type III: 2 m × 2 m × 2 m; type IV: 2.1 m × 2.1 m × 2.1 m).

**Figure 3 animals-15-00165-f003:**
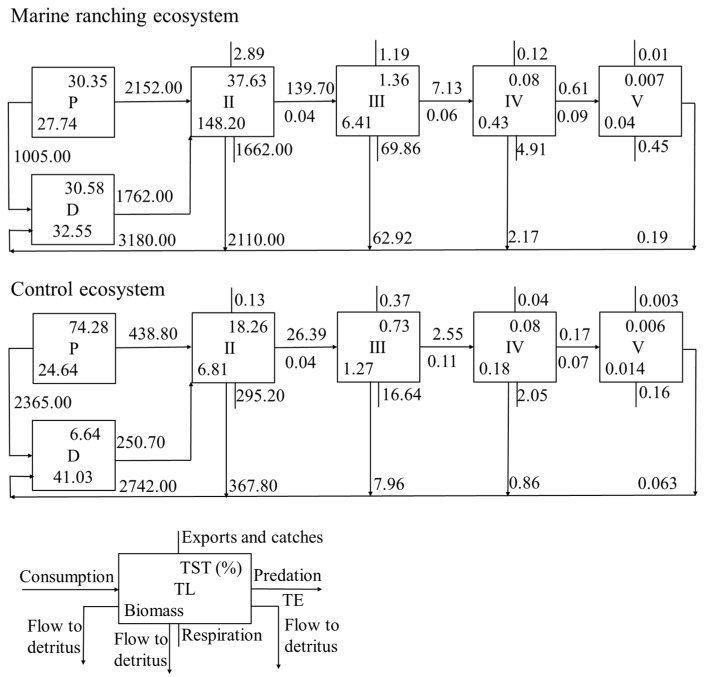
The energy flow (t/km^2^/a) among the different trophic levels in the marine ranching and control ecosystems, respectively (D: detritus; P: primary producers; TL: trophic level; TE: transfer efficiency; TST: total system throughput).

**Table 1 animals-15-00165-t001:** Selected indices and their weight assignment.

Overall Target	Criterion Layer	Indicator Layer	Weight
Ecosystem state	Ecosystem function	D/H	0.133
	0.4	A/C	0.133
		TE	0.133
	Food web structure	CI	0.114
	0.4	SOI	0.114
		FCI	0.114
		FML	0.057
	Ecosystem maturity	TPP/TR	0.1
	0.2	TPP/TB	0.05
		TB/TST	0.05

D/H: detritivory: herbivory ratio; TE: transfer efficiency; A/C: relative ascendancy; FCI: Finn’s cycling index; CI: connectance index; SOI: system omnivory index; FML: Finn’s mean path length; TPP/TR: total primary production/total respiration; TPP/TB: total primary production/total biomass; TB/TST: total biomass/total throughput.

**Table 2 animals-15-00165-t002:** The values of interval points across the different evaluation grades for indicators.

Indicator	Grades					Indicator Type
	Poor	Relatively Poor	Medium	Relatively Good	Good	
D/H	0.073	0.414	0.540	0.631	1.209	Positive
TE	2.920	6.800	9.400	11.500	13.236	Positive
A/C	0.150	0.264	0.302	0.339	0.368	Positive
CI	0.100	0.204	0.265	0.310	0.348	Positive
SOI	0.009	0.144	0.180	0.210	0.271	Positive
FCI	0.650	2.800	4.980	9.400	14.700	Positive
FML	1.206	2.336	2.568	3.193	4.000	Positive
TPP/TR	15.509	3.522	2.572	1.922	1.346	Negative
TPP/TB	132.000	48.656	31.964	17.338	9.110	Negative
TB/TST	0.003	0.008	0.011	0.019	0.030	Negative

The abbreviations are the same as in Table 1.

**Table 3 animals-15-00165-t003:** The TL, B, P/B, Q/B, EE, and Ui of each functional group in the marine ranching ecosystem.

Functional Group	TL	B (t/km^2^)	P/B (/a)	Q/B (/a)	EE	Ui
Pelagic fishes	2.70	0.11	0.60	16.29	0.76	0.20
Large and medium demersal fishes	3.42	0.18	1.16	12.82	0.77	0.20
Sparids	3.46	0.33	0.95	14.43	0.19	0.20
Leiognathidae	2.86	0.24	2.40	14.12	0.50	0.20
Small demersal fishes	3.22	0.18	1.76	13.60	0.72	0.20
Scorpaenidae	3.27	0.02	0.97	4.00	0.25	0.20
Gobiidae	3.22	0.07	2.32	12.96	0.86	0.20
Mantis shrimps	3.08	0.12	5.98	30.81	0.18	0.20
Large crabs	2.89	0.30	5.49	26.79	0.92	0.20
Other crabs	3.13	0.30	6.82	30.35	0.75	0.20
*Metapenaeopsis barbata*	2.33	0.52	7.89	27.61	0.31	0.20
Other shrimps	2.37	1.50	6.68	23.92	0.63	0.20
Cephalopods	3.28	0.07	3.81	14.83	0.80	0.20
Sea urchins	2.10	11.43	6.38	23.60	0.01	0.20
Gastropods	2.46	0.88	5.75	23.83	0.63	0.20
Barnacles	2.02	30.00	6.15	27.19	0.01	0.20
Oysters	2.02	21.76	4.61	20.93	0.07	0.40
Mussels	2.02	80.40	5.06	19.34	0.05	0.40
Other bivalves	2.02	1.58	6.20	23.46	0.74	0.40
Other benthos	2.34	0.93	6.10	21.95	0.61	0.35
Zooplankton	2.00	4.17	32.54	192.47	0.73	0.40
Phytoplankton	1.00	27.74	113.82		0.68	
Detritus	1.00	32.55			0.55	

TL: trophic level; B: biomass; P/B: production/biomass; Q/B: consumption/biomass; EE: ecotrophic efficiency; Ui: unassimilated consumption.

**Table 4 animals-15-00165-t004:** The TL, B, P/B, Q/B, EE, and Ui of each functional group in the control ecosystem.

Functional Group	TL	B (t/km^2^)	P/B (/a)	Q/B (/a)	EE	Ui
Pelagic fishes	2.70	0.06	0.60	16.29	0.73	0.20
Large and medium demersal fishes	3.35	0.18	1.30	15.18	0.78	0.20
Sparids	3.63	0.01	0.73	15.87	0.69	0.20
Leiognathidae	2.84	0.18	2.75	17.81	0.01	0.20
Small demersal fishes	3.16	0.12	1.83	12.75	0.61	0.20
Scorpaenidae	3.30	0.00	0.97	4.00	0.16	0.20
Gobiidae	3.16	0.07	2.30	13.92	0.41	0.20
Mantis shrimps	3.01	0.13	5.18	31.68	0.13	0.20
Large crabs	2.84	0.33	4.67	24.38	0.34	0.20
Other crabs	3.09	0.11	4.84	24.90	0.92	0.20
*Metapenaeopsis barbata*	2.31	0.06	7.40	28.04	0.91	0.20
Other shrimps	2.37	0.58	6.77	25.85	0.96	0.20
Cephalopods	3.25	0.07	4.58	15.82	0.36	0.20
Gastropods	2.42	0.26	5.70	23.80	0.65	0.20
Bivalves	2.01	2.86	5.65	23.77	0.95	0.40
Other benthos	2.08	0.11	6.10	21.95	0.89	0.35
Zooplankton	2.00	3.13	32.39	192.29	0.07	0.40
Phytoplankton	1.00	24.64	113.82		0.16	
Detritus	1.00	41.03			0.09	

The abbreviations are the same as in Table 3.

**Table 5 animals-15-00165-t005:** Ecosystem attributes of the marine ranching and control ecosystems.

Attribute	Marine Ranching Ecosystem	Control Ecosystem	Unit
Total system throughput	10,404.99	3777.48	t/km^2/^a
Total consumption	4064.63	720.83	t/km^2^/a
Total export	1422.78	0.54	t/km^2^/a
Total respiration	1737.31	314.03	t/km^2^/a
Total flow to detritus	3180.26	2742.08	t/km^2^/a
Total production	4098.69	2932.48	t/km^2^/a
Total primary production	3157.37	2804.18	t/km^2^/a
Total biomass	182.83	32.90	t/km^2^

**Table 6 animals-15-00165-t006:** The values of the ENA indicators in the marine ranching and control ecosystems.

Ecosystem	ENA Indicators									
	Ecosystem Function			Food Web Structure				Ecosystem Maturity		
	D/H	TE	A/C	CI	SOI	FCI	FML	TPP/TR	TPP/TB	TB/TST
Marine ranching	0.82	5.84	0.22	0.28	0.16	12.43	3.29	1.82	17.27	0.018
Control ecosystem	0.57	6.47	0.48	0.32	0.2	2.55	2.23	8.93	85.22	0.005

ENA: ecological network analysis indicator. The abbreviations of D/H, TE, A/C, CI, SOI, FCI, FML, TPP/TR, TPP/TB, and TB/TST are the same as in Table 1.

**Table 7 animals-15-00165-t007:** Evaluation results of the fuzzy synthetic evaluation for the status of marine ranching and control ecosystems.

Type	Ecosystem	Evaluation Grade of ENA Indices or Ecosystem Status				
		Poor	Relatively Poor	Medium	Relatively Good	Good
CI	Marine ranching ecosystem	0	0	0.67	0.33	0
	Control ecosystem	0	0	0	0.74	0.26
SOI	Marine ranching ecosystem	0	0.55	0.45	0	0
	control ecosystem	0	0	0.33	0.67	0
FCI	Marine ranching ecosystem	0	0	0	0.43	0.57
	control ecosystem	0.12	0.88	0	0	0
FML	Marine ranching ecosystem	0	0	0.12	0.88	0
	control ecosystem	0.09	0.91	0	0	0
D/H	Marine ranching ecosystem	0	0.32	0.68	0	0
	control ecosystem	0	0.4	0.6	0	0
A/C	Marine ranching ecosystem	0.42	0.58	0	0	0
	control ecosystem	0.03	0.97	0	0	0
TPP/TR	Marine ranching ecosystem	0	0	0	0.82	0.18
	control ecosystem	0.45	0.55	0	0	0
TPP/TB	Marine ranching ecosystem	0	0	0	0.99	0.01
	control ecosystem	0.44	0.56	0	0	0
TB/TST	Marine ranching ecosystem	0	0	0.13	0.88	0
	Control ecosystem	0.6	0.4	0	0	0
TE	Marine ranching ecosystem	0.25	0.75	0	0	0
	control ecosystem	0.09	0.91	0	0	0
Total ecosystem status	Marine ranching ecosystem	0.08	0.26	0.23	0.35	0.09
	Control ecosystem	0.14	0.54	0.11	0.17	0.03

The values in the table indicate the degree of membership for each indicator and the ecosystem within each evaluation grade. The highest value represents the status level of the indicators or ecosystem.

**Table 8 animals-15-00165-t008:** Carrying capacity of the marine ranching and control ecosystems.

Functional Group	Carrying Capacity (t/km^2^)	
	Marine Ranching	Control Ecosystem
Pelagic fishes	2.12	0.48
Large and medium demersal fishes	0.23	0.24
Sparids	0.38	0.027
Leiognathidae	0.9	0.26
Small demersal fishes	0.62	0.16
Scorpaenidae	0.62	0.11
Gobiidae	0.87	0.098
Mantis shrimps	0.34	0.12
Large crabs	0.85	0.40
Other crabs	0.86	0.135
*Metapenaeopsis barbata*	1.6	0.40
Other shrimps	3.1	0.95
Cephalopods	0.34	0.12
Gastropods	1.82	0.36
barnacles	90	/
Oysters	96	/
Mussels	163	/

**Table 9 animals-15-00165-t009:** The enhancing potential of each functional group in the marine ranch.

Category	Functional Group	Enhancing Potential (t/km^2^)
TL 3.0–3.5	Large and medium demersal fishes	0.05
	Sparids	0.05
	Small demersal fishes	0.44
	Scorpaenidae	0.60
	Gobiidae	0.80
	Cephalopods	0.27
	Mantis shrimps	0.17
	Other crabs	0.56
TL 2.5–3.0	Leiognathidae	0.52
	Pelagic fishes	2.01
	Large crabs	0.55
TL 2.0–2.5	Other shrimps	1.60
	*Metapenaeopsis barbata*	1.08
	Gastropods	0.94
	Barnacles	60.00
	Oysters	74.24
	Mussels	82.60
	Sea urchins	15.40

**Table 10 animals-15-00165-t010:** ENA indicators in marine ranching under different simulation scenarios.

Simulation Scenario	ENA Indicators									
	Ecosystem Function			Food Web Structure				Ecosystem Maturity		
	D/H	TE	A/C	CI	SOI	FCI	FML	TPP/TR	TPP/TB	TB/TST
Only MOB	0.74	5.25	0.27	0.28	0.16	20.12	3.97	1.18	10.89	0.02
MOB + fishing	/	/	/	/	/	/	/	/	/	/
Only crab	0.73	5.14	0.22	0.28	0.15	12.45	3.3	1.82	17.24	0.02
Crab + fishing	/	/	/	/	/	/	/	/	/	/
Only fish	0.73	5.45	0.22	0.28	0.13	12.44	3.29	1.81	17.22	0.02
Fish + fishing	/	/	/	/	/	/	/	/	/	/
MOB + crab	0.73	5.22	0.27	0.28	0.16	20.13	3.97	1.19	10.9	0.02
MOB + crab + fishing	0.73	5.68	0.27	0.28	0.16	20.14	3.97	1.19	10.9	0.02
MOB + fish	0.73	5.48	0.27	0.28	0.14	20.11	3.97	1.18	10.87	0.02
MOB + fish + fishing	/	/	/	/	/	/	/	/	/	/
Crab + fish	0.74	5.46	0.22	0.28	0.12	12.39	3.29	1.81	17.21	0.02
Crab + fish + fishing	/	/	/	/	/	/	/	/	/	/
MOB + crab + fish	0.74	5.44	0.27	0.28	0.14	20.12	3.97	1.19	10.88	0.02
MOB + crab + fish + fishing	0.74	5.89	0.27	0.28	0.14	20.13	3.97	1.19	10.88	0.02
Only fishing	/	/	/	/	/	/	/	/	/	/

Crab and fish represent the functional groups of large crabs and scorpaenidae, respectively. MOB: mussels, oysters, and barnacles.

**Table 11 animals-15-00165-t011:** Evaluation results of the fuzzy synthetic evaluation for the status of the marine ranch under different simulation scenarios.

Simulation Scenario	Poor	Relatively Poor	Medium	Relatively Good	Good
Only MOB	0.05	0.29	0.22	0.08	0.36
Only crab	0.10	0.26	0.20	0.29	0.15
Only fish	0.10	0.29	0.17	0.34	0.10
MOB + crab	0.05	0.28	0.16	0.08	0.36
MOB + crab + fishing	0.03	0.29	0.23	0.08	0.36
MOB + fish	0.04	0.34	0.17	0.08	0.36
Crab + fish	0.10	0.28	0.17	0.34	0.10
MOB + crab + fish	0.04	0.34	0.17	0.08	0.36
MOB + crab + fish + fishing	0.03	0.35	0.17	0.08	0.36

The values in the table indicate the degree of membership for the ecosystem within each evaluation grade. The highest value represents the status level of the ecosystem. MOB: mussels, oysters, and barnacles.

## Data Availability

The original contributions presented in the study are included in the article; further inquiries can be directed to the corresponding authors.

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
