# Peer review of "Evaluating Ecosystem Characteristics and Ecological Carrying Capacity for Marine Fauna Stock Enhancement Within a Marine Ranching System"

_animals, 2025, doi:10.3390/ani15020165_

Round 1

Reviewer 1 Report

Comments and Suggestions for Authors

This paper concerns an aspect of policy that is of particular importance in China. It is notable that the vast majority of the references listed are written in China, and a high proportion are in Chinese and thus will be inaccessible to most readers. As the authors explain in the introduction, the coastal waters of China are now either highly polluted by outflows from the land, or extensively occupied by aquaculture industries (which produce their own pollution and alter the environment). It is not surprising that ecosystems have been severely impacted and fishery yields have declined. This paper discusses how to maximise the productive capacity of coastal ecosystems for human use - an issue that may become much more important for other nations in future. Many countries however, would not view the development of “a comprehensive industrial development framework for marine ranching” as a key strategy for upgrading marine fisheries and safeguarding the marine environment.

The paper however has some serious shortcomings. The experimental design is not optimal, and while this cannot be fixed, the paper should acknowledge the issues involved in the discussion. The authors should revise the methods to  inform the reader adequately as to how the simulations were carried out, and they should consider the relation between population sizes and growth rates in determining the optimal population size and fishing pressure to produce maximum sustainable harvests, rather than focus on increasing target species to their apparent ecological carrying capacity.

A major problem for research such as this is that we have as yet a very incomplete understanding of how ecosystems work - even defining the boundaries of an ecosystem is difficult, and the classification of an ecosystem as “healthy” is usually based on knowledge that a set of interacting species appears to have persisted for a long time in a given environment. Obviously that basis for classification cannot be applied to “marine ranching ecosystems”. Rigorous scientific logic is crucial for a paper on this topic. Vague terms such as “maintaining ecological balance” and “biomass community structure” are not defined in this paper, but must be carefully defined if the methods to achieve outcomes are to be properly assessed.

In addition, the modelling is based on comparing two areas: an artificial reef area is compared to one “control area”. The problem here is that material and energy flows in any two areas will differ for many reasons. Good experimental design requires that the artificial reef area should be compared to a number of similar control areas, so that the variation between the control areas can be used to gauge the importance and nature of the differences between the artificial reef area and the other areas. Obviously this requires much more work, but the lack of several controls seriously reduces the reliability of any conclusions. In this case for example, there are extensive river systems that flow out to the north of the artificial reef area, and the control area is situated further to the south, further away from the land. Those river outflows are likely to produce important differences between the ranching and control areas that have little to do with the presence of the ranching structures and species. The authors should acknowledge this issue of one control area, and phrase their conclusions more cautiously. 

It is also not clear why the control area chosen was not a reef area, which would have similar habitats to the artificial reef area, and thus similar species and functional groups. I note that few conclusions are based on the comparison of the marine ranching and control areas. Using a reef area as a control (and preferably several reef areas) would have provided much more useful information.  

The problem of how to assess the carrying capacity of “marine ranching ecosystems” is addressed using EwE models, which involve simulations of the material and energy flows between parts of an ecosystem. Such models however, cannot address other issues, such as the effect of density in allowing diseases to spread through populations, which is a crucial and well understood issue in aquaculture farms, where density is almost always maximised. The limitations of using such models should be discussed in this paper.

Another problem with EwE models is that it is almost impossible to sample, measure and thus include all the interacting species in an ecosystem. In this case the structures at 9 m depth would have some degree of benthic algae cover that has not been included, so that primary producers are under-represented. This is another issue that should be acknowledged in the discussion. The paper should also make clear if the depth of the control area was the same as the marine ranching area, and what the bottom substratum is.

A more general problem is that the concept of an Ecological carrying capacity for a component population assumes that the system remains close to an equilibrium. As Connell and others have noted, many systems are subject to constant disturbance and are usually far from an equilibrium state. For many populations it seems probable that as its size increases, the chance of an event that reduces it (such as disease or unfavourable conditions) increases. Another way to view this is that populations at close to their “carrying capacity” are less stable. If marine ranching systems are to be managed, it would be better to maintain the key components of interest to humans at below their estimated carrying capacity, especially as the estimated level might exceed the actual capacity.  I note that the discussion in lines 572-576 goes some way to acknowledge the risks of boosting populations to an estimated carrying capacity, but more could be added.

In this paper (lines 301-4), the authors state: “setting the biomass of the stock enhancement group at ecological carrying capacity became one of the driving factors in establishing the Ecosim model. Additionally, fishing effort was also taken into account as a driving factor for model construction.” But as texts such as Beverton and Holt make clear, the optimal population size for a harvested resource (to obtain the maximum sustainable yield) is well below the ecological carrying capacity, so this does not make sense. If the ecosystem modelled included the fishers, was fishing effort included while the marine ranching functional groups were increased until just before the ecosystem became unbalanced? This method would not produce what would be normally regarded as an ‘ecological carrying capacity’. And did fishing effort change as the stocks increased? This would likely occur in the real world. The authors should explain clearly what they have simulated. The results also indicate that fishing effort was increased in some scenarios, apparently to some level regarded as the “carrying capacity” for fishing effort (although the methods do not make clear what was done). But unsurprisingly, many of these scenarios “with fishing” led to the elimination of some species targeted by fishers. It seems that the method of calculating a carrying capacity of fishing effort did not produce a sustainable “carrying capacity” when used in a simulation, which suggests a problem with the way the model was used.

The discussion suggests future research should determine optimal stocking densities based on mortality rates and growth characteristics of stock enhancement species. I suggest the authors should discuss the stock levels that would provide the best sustainable harvest yields, and consider how fishing effort could be managed to ensure the sustainable harvest laelel is not exceeded. Fisheries texts all discuss the relation between optimal stock size, stock productivity and harvest yield. At the level of the Ecological carrying capacity, the productivity of the stock is very low.

Specific comments:

Line 124: “SuiTable Atock enhancement” should be ‘Suitable stock enhancement’. A similar error appears in the discussion (line 613). 

Line 172: the whiskered velvet shrimp scientific name should be Metapenaeopsis barbata - in italics, with no capital for the species name.

Lines 192-194: The two statements here are contradictory. If the estimated EE was <1, then the estimated EE value could not have exceeded 1. If the second EE refers to whole trophic levels, or if Ecopath estimates EE internally, by modifying the initial values, this should be explained.

Line 214: the use of the weights in table 1 is not explained in the text.

Lines 266-7: The sentence “Preferably, groups with higher stock enhancement potential within each TL category were selected as stock enhancement groups.” does not make sense. Either these groups were selected or they were not. Perhaps the authors mean that these groups were selected unless enhancement technologies for them were not well developed.

Lines 281-285: In what sense is the formula in equation (1) empirical? What experiments or observations were used to derive it? Reference 70 provides no help. As the minimum TL for marine ranching is 2, this formula obviously sets vi to a top-down impact for all the marine ranching groups. Why?

Lines 306-310: This paragraph is so badly written as to be non-sensical. I am sure the authors do not mean that the three chosen stock enhancement groups - mussels, large crabs and scorpaenid fish - shared similar ecological characteristics, and were combined into one stock enhancement group. The acronym MOB suggests mussels, oysters and barnacles were combined into one functional group, although of these, only mussels were initially chosen for stock enhancement.

Lines 314-316: The meaning of stock enhancement with fishing and without fishing is not made clear. Does this mean that the fishing effort was increased in the scenarios “with fishing” and kept constant in those “without fishing”? or was fishing effort reduced to zero in those scenarios “without fishing”? Was there an “ecological carrying capacity” estimated for fishing effort, and the effort increased to this level? These issues need to be explained clearly for readers to understand the reported results.

Lines 421-4: As Mussels, oysters and barnacles are all enhanced by deploying more artificial reefs, and their combined carrying capacity was estimated, why are only mussels and oysters shown in Table 9?

Lines 430-432 and 456-440: In the case of stock enhancement of MOB + fishing, the text states that the large crab biomass declined to zero, indicating a collapse in the energy flow structure, but Table 10 shows fairly good values of ecosystem function. Presumably the text statement is an error.  In the case of MOB + Scorpaenidae + fishing, the large crab biomass again declined to zero, indicating a collapse in the energy flow structure. Yet in lines 438-440, the ecosystem status is stated to have improved to a “good” level. Presumably this is another error, as Table 10 shows no results for this intervention. I note that the discussion states that both MOB + fishing and MOB + Scorpaenidae + fishing did not elevate the ecosystem status.

Author Response

Reviewer #1

Q1 This paper concerns an aspect of policy that is of particular importance in China. It is notable that the vast majority of the references listed are written in China, and a high proportion are in Chinese and thus will be inaccessible to most readers. As the authors explain in the introduction, the coastal waters of China are now either highly polluted by outflows from the land, or extensively occupied by aquaculture industries (which produce their own pollution and alter the environment). It is not surprising that ecosystems have been severely impacted and fishery yields have declined. This paper discusses how to maximise the productive capacity of coastal ecosystems for human use - an issue that may become much more important for other nations in future. Many countries however, would not view the development of “a comprehensive industrial development framework for marine ranching” as a key strategy for upgrading marine fisheries and safeguarding the marine environment.

The paper however has some serious shortcomings. The experimental design is not optimal, and while this cannot be fixed, the paper should acknowledge the issues involved in the discussion. The authors should revise the methods to inform the reader adequately as to how the simulations were carried out, and they should consider the relation between population sizes and growth rates in determining the optimal population size and fishing pressure to produce maximum sustainable harvests, rather than focus on increasing target species to their apparent ecological carrying capacity.

Response: Thank you very much for your suggestions. We have revised the manuscript to address your concerns. In the Materials and Methods, and discussion section, we have elaborated on the selection issues with the control area (Line 190-196, 653-657), the limitations of the Ecopath model for assessing ecological carrying capacity (Line 682-699), and the risks and shortcomings associated with the simulation of maintaining the biomass of stock enhancement groups at the group’s carrying capacity (Line 682-715). These points are thoroughly discussed in the document. Additionally, we have provided a more detailed explanation of our simulation methods in the Materials and Methods section (Line 212-233, 330-340, 370-402). We also explore how our study contributes to understanding population size and fishing pressure required to produce maximum sustainable yields, discussing methods for achieving and managing these yields (Line 716-735). These details are provided in the text. We greatly appreciate your feedback, which has significantly helped to enhance the quality of our paper.

Q2 A major problem for research such as this is that we have as yet a very incomplete understanding of how ecosystems work - even defining the boundaries of an ecosystem is difficult, and the classification of an ecosystem as “healthy” is usually based on knowledge that a set of interacting species appears to have persisted for a long time in a given environment. Obviously that basis for classification cannot be applied to “marine ranching ecosystems”. Rigorous scientific logic is crucial for a paper on this topic. Vague terms such as “maintaining ecological balance” and “biomass community structure” are not defined in this paper, but must be carefully defined if the methods to achieve outcomes are to be properly assessed.

Response: Thank you for your comments and insights. We understand that fully comprehending how ecosystems function is indeed a significant challenge.

Defining the boundaries of marine ranching is currently a widely discussed issue in China. According to prior research on the impact range of artificial reefs, the Ministry of Agriculture and Rural Affairs of China has recommended that the area of artificial reefs within marine ranching should not exceed 30% of the total area. We have followed this guideline in determining the boundaries of our marine ranching study. While it is challenging to define the exact influence of individual artificial reefs, we believe that, the ecological impact of the artificial reef areas should not exceed the extent of the marine ranching area. This does not include the impact on migratory species, which, if considered, could result in a larger impact range.

 In our study, we applied ecological network analysis theory to assess the health of marine ranching ecosystems. If we were to only focus on the food production function of marine ranching, using health theories to evaluate the effectiveness of ranching construction and suggest optimizations, the results would be limited. To improve production efficiency, the ecosystem needs to remain in a phase of rapid development, where economic species grow quickly, and resource outputs are high. However, during this phase, the health level of the ecosystem may not be optimal. On the other hand, for long-term productivity, a healthy ecosystem is crucial, especially for marine ranching systems relying on natural productivity. Only with a well-established food web, good water quality and benthic environment can marine ranching continue to yield sustainable outputs. As a healthy and stable ecosystem is crucial for resisting human disturbances and climate change.

Given the heavy fishing pressure on China’s coastal waters over the past few decades, ecosystems have been severely impacted, with damaged food webs and dwindling fish stocks. In some areas, even the seabed has been left barren. This necessitates large-scale ecological restoration efforts, such as the deployment of artificial reefs, the establishment of seagrass beds, seaweed beds, and stocking efforts. From the perspective of ecological restoration, improving the health of marine ranching ecosystems is critical. This will not only enhance their functions, such as providing foraging, breeding, and nursery habitats, and preserving genetic resources, but also have a positive effect on the structure and function of the entire inshore ecosystem.

Overall, we believe that research on the energy flow, trophic structure, as well as the maturity, and stability of the marine ranching ecosystem is essential for improving ecosystem health. Furthermore, the theory of healthy ecosystems is applicable to the construction of marine ranching in China.

We appreciate your suggestion to better define ecological terms in the paper, and we have included more precise definitions of key terms, such as ecological balance and energy balance, in the revised manuscript to enhance the clarity and feasibility of the methodology. Please refer to the updated definitions in the paper for further details.

We sincerely thank you for prompting us to refine our definitions and clarify our methodology, which has undoubtedly strengthened the paper's scientific merit.

Q3 In addition, the modelling is based on comparing two areas: an artificial reef area is compared to one “control area”. The problem here is that material and energy flows in any two areas will differ for many reasons. Good experimental design requires that the artificial reef area should be compared to a number of similar control areas, so that the variation between the control areas can be used to gauge the importance and nature of the differences between the artificial reef area and the other areas. Obviously this requires much more work, but the lack of several controls seriously reduces the reliability of any conclusions. In this case for example, there are extensive river systems that flow out to the north of the artificial reef area, and the control area is situated further to the south, further away from the land. Those river outflows are likely to produce important differences between the ranching and control areas that have little to do with the presence of the ranching structures and species. The authors should acknowledge this issue of one control area, and phrase their conclusions more cautiously.

Response: Thank you for your comments. We acknowledge your concern regarding the comparison between the artificial reef area and a single control area, and appreciate the suggestion to include multiple control areas to enhance the reliability of our conclusions.

The control area to the south of the marine ranch was selected because its distance to the land is approximately equal to that of the marine ranch, both situated west of Guantou Ridge. Pre-construction site surveys, indicated that the ecological conditions, particularly water quality, and bottom substrate characteristics, are remarkably similar between the ranching and this southern area, thereby guiding our choice of control location. 

Indeed, as you correctly pointed out, the ranching area is closer to the northern shoreline, which introduces certain variabilities such as terrestrial inputs and hydrodynamic differences that might not be as pronounced in the control area. While it is challenging to find a perfectly matching control in field conditions, akin to controlled laboratory settings, we have addressed this limitation in the discussion section of our manuscript. We carefully phrased our conclusions to reflect the cautious interpretation of our findings, considering the possible influences and discrepancies brought about by the unique geographical settings of the control and ranching areas.

We have added a detailed explanation of these methodological choices and their implications in the discussion section, aiming to transparently convey the constraints and considerations that shaped our study.

Q4 It is also not clear why the control area chosen was not a reef area, which would have similar habitats to the artificial reef area, and thus similar species and functional groups. I note that few conclusions are based on the comparison of the marine ranching and control areas. Using a reef area as a control (and preferably several reef areas) would have provided much more useful information. 

Response: We chose a non-reef area as the control primarily to assess the effects of marine ranching construction. This decision was based on the ecological characteristics of the marine ranch and control area before construction, which were very similar, including distance from land, water quality, benthic environment, water depth, and biological community structure. To effectively evaluate the impact of the marine ranching construction compared to the pre-construction state, it was necessary to find an area that represented the baseline ecological characteristics of the marine ranching. Although it would have been ideal to use the pre-construction survey data as a baseline for comparison, however, the pre-construction survey data were not sufficient to support the establishment of a food web model. Additionally, using the pre-construction area as a control neglected the dynamics of the ecosystem during the construction period, leading to significant errors.

If we had used a natural reef ecosystem as a control, our evaluation would focus on the similarity between the post-construction artificial reef ecosystem and natural reef ecosystem. However, this would not allow us to compare with the baseline conditions before construction, preventing us from determining the actual improvements made by the ranching construction. Ideally, using both a natural reef area and an area representing the pre-construction baseline ecosystem as controls would enhance our study. This approach would allow for a more comprehensive quantification of the effects of marine ranching construction. By comparing these controls, we can more accurately assess the differences with both the natural reef and the baseline ecosystem. Whereas in the Beibu Gulf, the ecosystems of natural reef areas, such as the one around Weizhou Island, differ significantly from the marine ranching ecosystem in terms of water depth, distance from shore, and external nutrient inputs, these areas are not suitable to serve as control sites for this study. As a result, we ultimately selected a non-reef ecosystem located near the ranching area as our control. This area is ecologically similar to the ranching but minimally affected by the ranching construction.

Q5 The problem of how to assess the carrying capacity of “marine ranching ecosystems” is addressed using EwE models, which involve simulations of the material and energy flows between parts of an ecosystem. Such models however, cannot address other issues, such as the effect of density in allowing diseases to spread through populations, which is a crucial and well understood issue in aquaculture farms, where density is almost always maximised. The limitations of using such models should be discussed in this paper.

Response: Thank you for your comments. We have carefully addressed this concern in our revised manuscript. Specifically, we have clarified that the carrying capacity in this study was calculated using a static energy flow model, which represents a theoretical value derived from the perspective of an ecosystem's energy balance.

As noted in the revision, the model does not account for the dynamics of biological growth, migrations within different functional groups, or fluctuations in environmental factors. Furthermore, we acknowledge that in ecosystems where the biomass or density of a single species significantly increases, additional issues such as disease outbreaks and interspecies competition may arise. These factors, which are crucial in aquaculture systems, are not considered by the model and may lead to overestimation of the actual ecological carrying capacity.

Q6 Another problem with EwE models is that it is almost impossible to sample, measure and thus include all the interacting species in an ecosystem. In this case the structures at 9 m depth would have some degree of benthic algae cover that has not been included, so that primary producers are under-represented. This is another issue that should be acknowledged in the discussion. The paper should also make clear if the depth of the control area was the same as the marine ranching area, and what the bottom substratum is.

Response: Thank you for your comments and suggestions. Regarding the survey and sampling of different categories of organisms, further detailed explanations are provided in the Materials and Methods section. We acknowledge the issue of potential underrepresentation of primary producers in the Ecopath model. In our detailed biological resource survey of the study area, we did not observe the presence of large macrophytes. However, we recognize that benthic microalgae, which can be abundant in shallow coastal waters, should still have relatively high primary productivity under these conditions (Cahoon, 2002). Unfortunately, the Ecopath model we constructed did not incorporate the primary productivity of benthic microalgae, which has led to an underestimation of the total ecosystem primary productivity. We have updated the discussion section to acknowledge this limitation and its potential impact on our model’s results.

Regarding your suggestion about clarifying the depth and bottom substratum of the control area, we confirm that the depth of the control area is approximately 9 meters, similar to the marine ranching area. The bottom substratum in the control area is sandy silt. We have added this information to the revised manuscript.

Q7 A more general problem is that the concept of an Ecological carrying capacity for a component population assumes that the system remains close to an equilibrium. As Connell and others have noted, many systems are subject to constant disturbance and are usually far from an equilibrium state. For many populations it seems probable that as its size increases, the chance of an event that reduces it (such as disease or unfavourable conditions) increases. Another way to view this is that populations at close to their “carrying capacity” are less stable. If marine ranching systems are to be managed, it would be better to maintain the key components of interest to humans at below their estimated carrying capacity, especially as the estimated level might exceed the actual capacity. I note that the discussion in lines 572-576 goes some way to acknowledge the risks of boosting populations to an estimated carrying capacity, but more could be added.

Response: Thank you for your comments. In response to your discussion on the increase in mortality rates or the decrease in environmental suitability as biological populations grow within ecosystems, we have further reviewed the related articles, and have added discussions on these aspects to our paper. Additionally, regarding the increased instability of ecosystems as biological populations approach their carrying capacities, we have also expanded our discussion in the manuscript. The specific content added is: as ecosystems approach their carrying capacities, constraints on resources such as prey and habitat space become more pronounced, potentially diminishing the resilience of these systems. These increased limitations may heighten the ecosystem's sensitivity to disturbances, leading to greater population volatility and instability. Under such conditions, even minor external disturbances or internal fluctuations could push the ecosystem into a state of energy imbalance, significantly elevating ecological risks.

Q8 In this paper (lines 301-4), the authors state: “setting the biomass of the stock enhancement group at ecological carrying capacity became one of the driving factors in establishing the Ecosim model. Additionally, fishing effort was also taken into account as a driving factor for model construction.” But as texts such as Beverton and Holt make clear, the optimal population size for a harvested resource (to obtain the maximum sustainable yield) is well below the ecological carrying capacity, so this does not make sense. If the ecosystem modelled included the fishers, was fishing effort included while the marine ranching functional groups were increased until just before the ecosystem became unbalanced? This method would not produce what would be normally regarded as an ‘ecological carrying capacity’. And did fishing effort change as the stocks increased? This would likely occur in the real world. The authors should explain clearly what they have simulated. The results also indicate that fishing effort was increased in some scenarios, apparently to some level regarded as the “carrying capacity” for fishing effort (although the methods do not make clear what was done). But unsurprisingly, many of these scenarios “with fishing” led to the elimination of some species targeted by fishers. It seems that the method of calculating a carrying capacity of fishing effort did not produce a sustainable “carrying capacity” when used in a simulation, which suggests a problem with the way the model was used.

Response: Thank you for your comments. In our study, we intentionally set the biomass of the stock enhancement group at the ecological carrying capacity to explore the implications of maximizing biomass within the ecosystem's dynamic constraints. The concept of ecological carrying capacity is pivotal as it represents the upper limit to which a functional group’s biomass can be increased before destabilizing the ecosystem. This threshold provides crucial insights for managers, offering guidance on potential risks and management strategies to avoid ecosystem collapse.

Our simulations aimed to observe the ecosystem's response when the biomass of a functional group reaches this critical state. Even the Ecopath model indicated that the ecosystem structure did not collapse at this point, our simulations suggested that increasing biomass to carrying capacity could lead to a gradual decrease in other groups’ biomasses, potentially leading to collapses. This serves as a warning that managing stock enhancement towards such high biomass levels may require adjustments to avoid detrimental effects on the ecosystem.

Regarding the maximum sustainable yield (MSY), our results indeed provide essential insights. According to the MSY theory (Schaefe, 1954), the MSY often occurs at about half the carrying capacity, offering a sustainable fishing benchmark. This critical information is aimed to assist managers in establishing sustainable harvest levels. However, our focus was not solely on achieving MSY but rather on evaluating the impacts of surpassing ecological carrying capacity through enhancement activities, which is often the case in many of China’s marine ranching operations. We sought to simulate potential risks associated with these common practices to inform and guide management decisions more effectively.

In our paper, regarding the simulation of fishing effort, it was maintained at a constant level throughout the simulation process. We have elaborated on the fishing strategy simulation in the manuscript following your recommendations. We did not simulate scenarios with varying fishing efforts for a couple of reasons. First, predicting future changes in fishing effort is particularly challenging. Second, after more than four years of development, the marine ranching ecosystem has reached a relatively stable state. Under these conditions, the current level of fishing is representative and, in the marine ranches we studied, the annual harvest amount is controllable by the ranching companies. Therefore, we based our predictions on a constant fishing effort, which provides actionable insights for the company's management. The company can then use this information to adjust future fishing levels as needed.

We hope that these explanations clarify our approach and address your concerns regarding our methodology.

Reference:

Schaefer MB. Some aspects of the dynamics of populations important to themanagement ofcommercial marine fisheries. Bull. Inter-Am. Trop. Tuna Comm.1. 25.56 (1954).

Q9 The discussion suggests future research should determine optimal stocking densities based on mortality rates and growth characteristics of stock enhancement species. I suggest the authors should discuss the stock levels that would provide the best sustainable harvest yields, and consider how fishing effort could be managed to ensure the sustainable harvest level is not exceeded. Fisheries texts all discuss the relation between optimal stock size, stock productivity and harvest yield. At the level of the Ecological carrying capacity, the productivity of the stock is very low.

Response: Thank you very much for your careful review and valuable feedback on our manuscript. We have addressed these points in the revised manuscript, and we would like to provide the following clarifications:

On the relationship between MSY and ecological carrying capacity: According to the MSY theory, the MSY often occurs at about half the carrying capacity. And the carrying capacity is obtained based on isolated populations growing characteristics according to logistic growth models. The carrying capacity derived from this approach may be higher than the ecological carrying capacity estimated in this study, but lower than the production carrying capacity, which is achieved at maximum production levels. We have included this in the revised manuscript, where we discuss that maximum sustainable production may fall between half of the ecological carrying capacity and half of the production carrying capacity. This range represents the point at which species would achieve the maximum sustainable harvest.

On managing fishing efforts to ensure sustainable harvest levels: In response to your suggestion, we have added a section in the revised manuscript discussing how fishing efforts can be managed to avoid exceeding the maximum sustainable yield. Specifically, we address species such as Sparids, and large and medium demersal fishes, whose biomass is approaching their carrying capacity, thus eliminating the need for stock enhancement. In contrast, species such as large crabs, Scorpaenidae, and oysters still maintain a biomass well below half of their carrying capacity, indicating considerable potential for stock enhancement. To mitigate the risks associated with such enhancement efforts, it would be prudent to manage the biomass of these groups at approximately half of their ecological carrying capacity.

Q10 Line 124: “SuiTable Atock enhancement” should be ‘Suitable stock enhancement’. A similar error appears in the discussion (line 613).

Response: the “suiTable Apecies for stock enhancement” has been changed to “suitable species for stock enhancement” in line 613, and the “SuiTable Atock enhancement” has been changed to “Suitable stock enhancement” in the revised manuscript.

Q11 Line 172: the whiskered velvet shrimp scientific name should be Metapenaeopsis barbata - in italics, with no capital for the species name.

Response: The “Metapenaeopsis Barbata” has been changed to “Metapenaeopsis barbata” in the revised manuscript.

Q12 Lines 192-194: The two statements here are contradictory. If the estimated EE was <1, then the estimated EE value could not have exceeded 1. If the second EE refers to whole trophic levels, or if Ecopath estimates EE internally, by modifying the initial values, this should be explained.

Response: Thank you for your comment. The intention of our original statement was to explain that we used the EE value of each functional group less than 1 as the standard for model calibration. If the estimated EE exceeded 1, indicating an imbalance between consumed and produced biomass, we made small adjustments to the diet composition of the consumer groups (with each change not exceeding 0.05) to bring the EE value below 1.

We have made further revisions to the relevant content in the text as follows: “We used the estimated ecotrophic efficiency (EE) value of each functional group (which should be <1) as the primary criterion for model calibration. If the estimated EE exceeded 1, indicating that the consumed biomass surpassed the produced biomass, we incrementally adjusted the diet composition of each consumer group, with each adjustment not exceeding 0.05, to reduce the EE value below 1”.   

Q13 Lines 266-7: The sentence “Preferably, groups with higher stock enhancement potential within each TL category were selected as stock enhancement groups.” does not make sense. Either these groups were selected or they were not. Perhaps the authors mean that these groups were selected unless enhancement technologies for them were not well developed.

Response: Thank you for your helpful comment; we did not state this clearly. The original intention was to select groups with high stock enhancement potential within each TL category, provided that the enhancement technologies, such as seedling breeding and larval releasing, were sufficiently developed. Groups with high enhancement potential but underdeveloped enhancement technologies were not selected as stock enhancement groups.

We have made further revisions to the relevant content in the text as follows: “Within each TL category, groups with higher stock enhancement potential were identified as candidates for stock enhancement. However, the maturity of relevant enhancement technologies, such as seedling breeding and larval releasing, was also considered critical for the final selection. Groups with high enhancement potential but underdeveloped enhancement technologies were excluded from the stock enhancement groups, and only those groups with both high enhancement potential and mature enhancement technologies were selected”.

Q14 Lines 281-285: In what sense is the formula in equation (1) empirical? What experiments or observations were used to derive it? Reference 70 provides no help. As the minimum TL for marine ranching is 2, this formula obviously sets vi to a top-down impact for all the marine ranching groups. Why?

Response:

Thank you for your questions. Equation (1) is empirical because it is derived directly from observational data and refined through rigorous model testing, primarily drawing on foundational works such as those by Cheung (2001) and Cheung et al. (2002). Cheung (2001) and Cheung et al. (2002) consistently demonstrated in their research that the model’s vulnerability values are linearly related to the trophic levels of each functional group. This is based on the general theory that species higher in the food web tend to be more frequently away from their shelters, thereby being in a more “vulnerable” state. According to Christensen et al. (2005), vulnerabilities represent the degree to which an increase in predator biomass will result in increased predation mortality for a given prey. Using this hypothesis, Cheung (2001) set the vulnerability indices in the construction of the Hong Kong Ecosim model, with trophic level data derived from Ecopath model calculations. The model successfully captured trends in biomass and fishing volumes from 1950 to 1998 in Hong Kong waters, including a noticeable decline in the average trophic level of fished species, indicating a positive linear relationship between model vulnerability values and trophic levels. This empirical relationship was further validated by fitting linear models to the trophic levels and final vulnerability indices of each functional group, leading to the formulation of this equation. This formula has been successfully applied in multiple papers (Buchary et al., 2003; Chen et al., 2008; Kluger et al., 2015; Bacalso et al., 2016). Overall, we believe that using this formula for our research in this paper is reasonable.

Regarding your last question: “As the minimum TL for marine ranching is 2, this formula obviously sets vi to a top-down impact for all the marine ranching groups. Why?”. I should clarify that in versions of the EwE model prior to 5.0, the Ecosim model's vulnerability values were set between 0 and 1, with 0.0–0.3 representing a bottom-up control, 0.3 indicating mixed control, and 0.3–1.0 indicating a top-down impact. However, in versions 5.0 and later of EwE, the model functionality has been improved; the range of vulnerability values was changed to 1 to infinity. In these newer versions, a vulnerability value closer to 1 indicates a strong top-down effect, and higher values indicate a more pronounced bottom-up effect. The equation (2) in our paper translates the original vi​ values, which range from 0 to 1 (suitable for earlier versions of the EwE model), into vnew, which range from 1 to infinity (suitable for the improved versions of the EwE model) (Kluger et al., 2015). As stated in the text: vi was then transformed to derive vnew ​for Ecosim input, which ranged from 1 to ∞ (The version of EwE used in this study is 6.7). Therefore, based on the calculations using Equation 2, when the trophic level of the species is 2, its vi is 0.35, and vnew​ is approximately 6.4, which may still represent a mixed control effect.

References:

Bacalso R T M, Wolff M, Rosales R M, et al. Effort reallocation of illegal fishing operations: A profitable scenario for the municipal fisheries of Danajon Bank, Central Philippines[J]. Ecological modelling, 2016, 331: 5-16.

Buchary E A, Alder J, Nurhakim S, et al. The use of ecosystem-based modelling to investigate multi-species management strategies for capture fisheries in the Bali Strait, Indonesia[J]. Fisheries Centre Research Reports, 2002, 10(2): 24.

Chen Z, Qiu Y, Jia X, et al. Simulating fisheries management options for the Beibu Gulf by means of an ecological modelling optimization routine[J]. Fisheries Research, 2008, 89(3): 257-265.

Christensen V, Walters C J, Pauly D. Ecopath with Ecosim: a user’s guide[J]. Fisheries Centre, University of British Columbia, Vancouver, 2005, 154: 31.

Kluger L C, Taylor M H, Mendo J, et al. Carrying capacity simulations as a tool for ecosystem-based management of a scallop aquaculture system[J]. Ecological modelling, 2016, 331: 44-55.

Q15 Lines 306-310: This paragraph is so badly written as to be non-sensical. I am sure the authors do not mean that the three chosen stock enhancement groups - mussels, large crabs and scorpaenid fish - shared similar ecological characteristics, and were combined into one stock enhancement group. The acronym MOB suggests mussels, oysters and barnacles were combined into one functional group, although of these, only mussels were initially chosen for stock enhancement.

Response: Thank you for your comments. We realize that the original paragraph was unclear and may have caused confusion. Specifically, only oysters and mussels were mentioned in this paragraph. We initially selected mussels as the target group for enhancement, and the method involved constructing artificial reefs to provide a suitable habitat for mussels. However, once the reefs were deployed, oysters and barnacles, in addition to mussels, grew on them, as these three groups share similar ecological characteristics. As a result, the groups being enhanced are these three. To clarify, large crabs and scorpaenid fish were not part of the MOB group. To better simulate the effect of this enhancement method, we included mussels, oysters, and barnacles (MOB) in the enhancement simulation and combined them into a single stock enhancement group. We have revised the original text to reflect this more clearly and avoid any misunderstanding.

Q16 Lines 314-316: The meaning of stock enhancement with fishing and without fishing is not made clear. Does this mean that the fishing effort was increased in the scenarios “with fishing” and kept constant in those “without fishing”? or was fishing effort reduced to zero in those scenarios “without fishing”? Was there an “ecological carrying capacity” estimated for fishing effort, and the effort increased to this level? These issues need to be explained clearly for readers to understand the reported results.

Response: Thank you for your comments. Concerning the setup of the “with fishing activity” and “without fishing activity” scenarios, we acknowledge that the original manuscript did not provide sufficient explanation, which may have caused some confusion. Here, we provide a detailed clarification of these scenarios:

Among the seven stock enhancement strategies we set, each strategy includes both “with fishing” and “without fishing” scenarios. Additionally, we also established a scenario involving only fishing, with no stock enhancement, resulting in a total of eight scenarios involving fishing activity. In these eight scenarios, the fishing effort is based on the marine ranching fishery catch levels provided by the Qinzhou Agriculture and Rural Bureau of Guangxi Zhuang Autonomous Region in 2023, combined with the stock enhancement activities, to simulate future changes in the ecosystem. In the fishing-included scenarios, the fishing effort remains constant throughout the simulation period. This approach was adopted because the marine ranching ecosystem is already in a relatively stable state. By maintaining a constant catch level and comparing it with the initial model, we aim to explore how different strategies optimize the ecosystem.

In the “without fishing” scenarios, all fishing activities are excluded, meaning that the fishing effort is set to zero. This allows us to simulate the effects of stock enhancement in the absence of fishing interference and predict the future impacts of the seven stock enhancement strategies on the ecosystem.

Regarding the “ecological carrying capacity” estimated for fishing effort, i.e., the MSY, this has not been directly estimated in our study. We only set a fixed level of fishing effort in the “with fishing activity” scenarios, assuming that this catch level represents a plausible fishing intensity. Although the MSY in the ranching ecosystem was not explicitly evaluated, we can estimate the carrying capacity of important economic species in the ecosystem for a future year using the Ecopath and Ecosim models. According to the MSY theory, the MSY often occurs at about half the carrying capacity, and the carrying capacity is obtained based on isolated populations growing characteristics according to logistic growth models. The carrying capacity derived from this approach may be higher than the ecological carrying capacity estimated in this study, but lower than the production carrying capacity, which is achieved at maximum production levels. The maximum sustainable production may fall between half of the ecological carrying capacity and half of the production carrying capacity. To achieve the MSY for economically important groups in marine ranching, it may be beneficial to implement appropriate harvesting activities for groups such as Sparids, and large and medium demersal fishes, whose biomass is approaching their carrying capacity, thus eliminating the need for stock enhancement. In contrast, species such as large crabs, Scorpaenidae, and oysters still maintain a biomass well below half of their carrying capacity, indicating considerable potential for stock enhancement. To mitigate the risks associated with such enhancement efforts, it would be prudent to manage the biomass of these groups at approximately half of their ecological carrying capacity.

Q17 Lines 421-4: As Mussels, oysters and barnacles are all enhanced by deploying more artificial reefs, and their combined carrying capacity was estimated, why are only mussels and oysters shown in Table 9?

Response: According to the reviewer’s suggestion, the carrying capacity of barnacles has been added in Table 8, and the enhancing potential of barnacles has also been added in Table 9 in the text.

Q18 Lines 430-432 and 456-440: In the case of stock enhancement of MOB + fishing, the text states that the large crab biomass declined to zero, indicating a collapse in the energy flow structure, but Table 10 shows fairly good values of ecosystem function. Presumably the text statement is an error.  In the case of MOB + Scorpaenidae + fishing, the large crab biomass again declined to zero, indicating a collapse in the energy flow structure. Yet in lines 438-440, the ecosystem status is stated to have improved to a “good” level. Presumably this is another error, as Table 10 shows no results for this intervention. I note that the discussion states that both MOB + fishing and MOB + Scorpaenidae + fishing did not elevate the ecosystem status.

Response: Thank you for your comments, which indeed identified two errors in our manuscript. We have made the following corrections in the text: In Table 10, we have removed the model parameters for the MOB + fishing scenario, replacing them with slash marks; and in line 438, we have deleted the MOB + scorpaenidae + fishing scenario. Furthermore, we have conducted a thorough review of the data and results described throughout the manuscript to ensure that such errors do not occur again.

Reviewer 2 Report

Comments and Suggestions for Authors

With increasing focus on sustainable aquaculture practices, this study aimed to evaluate the structural and functional dynamics of marine ranching compared to a nearby control ecosystem. They developed two energy flow models and an evaluation index system that allowed a comprehensive assessment of the carrying capacity and ecological health of the ecosystems.

The authors found that the marine ranching ecosystem had significant advantages over the control ecosystem in several key areas. The authors rated the ranching ecosystem as "relatively good". In addition, the study identified mussels, large crabs and Scorpaenidae as key species for stock enhancement efforts, highlighting the potential for these organisms to contribute significantly to the biomass and stability of the ranching ecosystem.

Overall, the paper is well written and well illustrated.

The authors have used statistical methods to process the data.

At the same time, there are some issues that need to be considered and addressed to improve the clarity of this manuscript.

Introduction

L 29: Here the authors mention 'inshore waters'. They should clarify this term as different authors consider this area of the sea in different ways.

L 66: The authors mention 'climate change' as a human activity. They should rephrase the text as climate change is not.

L 75: The authors should give a clear explanation of how marine ranching differs from related activities such as aquaculture and farming. How do the strategies for marine ranching in China compare with successful practices in other countries?

L 103: The authors mention risks but do not explain them. They should provide a description of the specific ecological and economic risks associated with relying on empirical knowledge rather than scientific rigour in marine ranching projects.

Materials and methods

L 143: The authors should briefly describe the ecological objectives of establishing the Jinggong marine ranching project at this specific site.

L 159: It is unclear how the authors surveyed the fish community. They should describe in detail any methods used to determine abundance and biomass. How many samples were analysed for each group? What were the main pelagic fishes?

L 170: The authors mention 23 functional groups, whereas Table S1 lists 24 groups.

L 260: The authors should mention how the study accounts for the variability of the different functional groups, especially their responses to environmental changes.

Results.

L 403: Pelagic fishes have been shown to have the highest carrying capacity among fish species. Which fish species do the authors mention here?

L 441: The authors mention 'a significant increase'. What statistical analysis was carried out to reach this conclusion?

Discussion.

L 457: The authors should specify what type of ranching is being considered in the area. What is the target organism for which sea ranching has been implemented in the area?

L 461-462: The authors should explain more clearly how the use of artificial reefs affects nutrient dynamics within the benthic environment. They should discuss the biochemical processes involved and their ecological implications.

L 486: The authors should discuss approaches that could be taken to improve TE between higher trophic levels in the marine ranching ecosystem. The authors should suggest potential management interventions or research directions aimed at improving energy transfer efficiency.

L 494: The authors should be more explicit about how external disturbances disrupt the self-organisation processes within the study area. For example, they should consider climate change or pollution and their effects on ecosystem dynamics.

L 503: It is unclear which species are overfished in the study area.

L 527-528: The statement 'While zooplankton can quickly respond to ecosystem disturbances, bivalves  exhibit  slower  responses, thus  creating  a  pathway for  slow  energy  flow' is controversial, as mussels can also respond quickly to disturbances.

Author Response

Reviewer #2

Comments and Suggestions:

With increasing focus on sustainable aquaculture practices, this study aimed to evaluate the structural and functional dynamics of marine ranching compared to a nearby control ecosystem. They developed two energy flow models and an evaluation index system that allowed a comprehensive assessment of the carrying capacity and ecological health of the ecosystems.

The authors found that the marine ranching ecosystem had significant advantages over the control ecosystem in several key areas. The authors rated the ranching ecosystem as "relatively good". In addition, the study identified mussels, large crabs and Scorpaenidae as key species for stock enhancement efforts, highlighting the potential for these organisms to contribute significantly to the biomass and stability of the ranching ecosystem.

Overall, the paper is well written and illustrated. The authors have used statistical methods to process the data. At the same time, there are some issues that need to be considered and addressed to improve the clarity of this manuscript.

Q1 L 29: Here the authors mention 'inshore waters'. They should clarify this term as different authors consider this area of the sea in different ways.

Response: Thank you for your comment. The term “inshore waters” as used in our manuscript, aligns with the definition of the Exclusive Economic Zone (EEZ), extending up to 200 nautical miles from the coastal baseline of a country. This definition is consistent with the United Nations Convention on the Law of the Sea (UNCLOS), which outlines the EEZ as a zone where a sovereign state has special rights over the exploration and use of marine resources. This encompasses both the water column and the seabed up to this limit, providing a clear and internationally recognized framework for our discussion of ecological services provided by inshore water areas.

Q2 L 66: The authors mention 'climate change' as a human activity. They should rephrase the text as climate change is not.

Response: We have revised the statement in the manuscript to more accurately reflect the relationship between human activities and climate change. The revised sentence as: “However, in recent years, human activities such as industrialization, urbanization, and agricultural expansion, along with the impacts of climate change, have severely degraded China's inshore habitats and fishery resources.”

Q3 L 75: The authors should give a clear explanation of how marine ranching differs from related activities such as aquaculture and farming. How do the strategies for marine ranching in China compare with successful practices in other countries?

Response: Thank you for your comments. We have revised the manuscript to address your concerns.

We have clarified that marine ranching differs from aquaculture and traditional farming primarily in its emphasis on ecosystem restoration and the enhancement of natural marine resources, rather than the intensive cultivation of species in controlled environments. While aquaculture typically involves the farming of specific species within enclosed systems (e.g., fish farms), marine ranching seeks to enhance marine ecosystems through practices such as artificial reef deployment, seaweed bed restoration, and hatchery release. These methods aim to restore ecological environment, improve biodiversity, and support the sustainable management of marine resources. Thus, marine ranching serves a broader ecological function than aquaculture, which is generally more focused on production.

Moreover, unlike stock enhancement through hatchery release alone, marine ranching places a greater emphasis on habitat restoration and comprehensive resource management. This integrated approach is designed to increase the survival and recapture rates of targeted species, thereby contributing to the long-term sustainability and resilience of marine ecosystems.

We have also expanded the manuscript to include a comparison of China's marine ranching strategies with those in other countries. As noted in the revised manuscript, marine ranching in China shares similarities with practices in countries such as Japan, South Korea, the United States, and Australia, particularly in the use of habitat restoration method and hatchery release, to enhance marine biodiversity and fishery resources. However, the key differences lie in the industrial models and approaches, which vary based on national contexts and priorities.

Q4 L 103: The authors mention risks but do not explain them. They should provide a description of the specific ecological and economic risks associated with relying on empirical knowledge rather than scientific rigour in marine ranching projects.

Response: We have addressed the request for a more detailed explanation of the ecological and economic risks in the text.

Q5 L 143: The authors should briefly describe the ecological objectives of establishing the Jinggong marine ranching project at this specific site.

Response: The ecological objectives of establishing the Jinggong marine ranching project were provided in the text.

Q6 L 159: It is unclear how the authors surveyed the fish community. They should describe in detail any methods used to determine abundance and biomass. How many samples were analysed for each group? What were the main pelagic fishes?

Response: We have detailed the methods used to survey fish abundance and biomass in the article. Due to differences in environmental conditions between reef areas and sandy silt bottom substrates, different survey methods were used.

In sandy silt substrate areas, the biomass of fish was estimated using the swept-area method. The hauling speed was approximately 2.5 knots, with an inner mesh size of 3.5 cm and a mouth width of 6 m. Each trawl lasted for 30 minutes. There were three survey sites in the marine ranching area and nine in the control area.

In reef areas, the fish survey was conducted using a transect method combined with cage net sampling. Divers performed transect-based swimming video recordings after descending into the water. The duration of each video at the stations was 15 minutes, with an average sweeping area of 18 square meters per station. Biomass of each fish species was calculated by multiplying the number of each fish species observed by the average weight derived from the cage net, divided by the swept area of the video recording. Six stations were set in the reef area. Additionally, we have detailed the survey methods for other swimming organisms and benthic animals.

Regarding the main pelagic fishes, as shown in Table A1, the primary species include Konosirus punctatus, Thryssa dussumieri, and Trachinotus ovatus.

These changes improve the grammatical correctness and clarity of your response

Q7 L 170: The authors mention 23 functional groups, whereas Table S1 lists 24 groups.

Response: Thank you for your comments. There was a small error in the numbering in Table A, which has now been corrected. The table now contains a total of 23 functional groups.

Q8 L 260: The authors should mention how the study accounts for the variability of the different functional groups, especially their responses to environmental changes.

Response: The study accounts for the variability of different functional groups by incrementally adjusting their biomasses during the calculation of ecological carrying capacity. This method is adapted from Wolff (1994), Jiang & Gibbs (2005), Byron et al. (2011a, 2011b), and Xu et al. (2011). Based on this approach, when evaluating the ecological carrying capacity of a specific functional group, the biomass of that group is modified independently, and the system is allowed to reach a state of balance or imbalance based on the specific biomass levels of that group. As the biomass of this functional group is gradually increased until the ecosystem’s energy balance reaches a critical point of collapse, no other parameters in the model are altered. At this point, the biomass of the functional group is considered its ecological carrying capacity.

Thus, the method does not explicitly model inter-functional group variability (such as differences in growth rates). However, the biomass adjustments inherently reflect the ecological relationships within the system. We acknowledge that changes in the biomass of one functional group may lead to changes in the parameters of other groups. In future work, we aim to develop more detailed models that incorporate the dynamic interactions and variabilities of the functional groups, thereby improving the accuracy of ecological carrying capacity assessments.

References:

Byron C, Link J, Costa-Pierce B, et al. Calculating ecological carrying capacity of shellfish aquaculture using mass-balance modeling: Narragansett Bay, Rhode Island[J]. Ecological Modelling, 2011a, 222(10): 1743-1755.

Byron C, Link J, Costa-Pierce B, et al. Modeling ecological carrying capacity of shellfish aquaculture in highly flushed temperate lagoons[J]. Aquaculture, 2011b, 314(1-4): 87-99.

Jiang W, Gibbs M T. Predicting the carrying capacity of bivalve shellfish culture using a steady, linear food web model[J]. Aquaculture, 2005, 244(1-4): 171-185.

Wolff M. A trophic model for Tongoy Bay—a system exposed to suspended scallop culture (Northern Chile)[J]. Journal of Experimental Marine Biology and Ecology, 1994, 182(2): 149-168.

Xu S, Chen Z, Li C, et al. Assessing the carrying capacity of tilapia in an intertidal mangrove-based polyculture system of Pearl River Delta, China[J]. Ecological Modelling, 2011, 222(3): 846-856.

Q9 L 403: Pelagic fishes have been shown to have the highest carrying capacity among fish species. Which fish species do the authors mention here?

Response: In this study, as shown in Table A1, the pelagic fishes group is composed of Konosirus punctatus, Thryssa dussumieri, and Trachinotus ovatus. According to the principles for setting up functional groups, a functional group can consist of ecologically related species, a single species, or a single size/age group of a species (Christensen et al., 2005). Since these three fish species share similar ecological characteristics, such as diets and habitat layers, they were combined into a single functional group. This approach enhances the clarity of the overall ecosystem structure and its functional characteristics, while maintaining simplicity in the ecosystem model.

Reference: Christensen V, Walters C J, Pauly D. Ecopath with Ecosim: a user’s guide[J]. Fisheries Centre, University of British Columbia, Vancouver, 2005, 154: 31.

Q10 L 441: The authors mention 'a significant increase'. What statistical analysis was carried out to reach this conclusion?

Response: Thank you for your comments. Regarding your query about the “significant increase”, we acknowledge that the original phrasing in our manuscript may have led to misunderstandings. Indeed, the decreases in TPP/TR and TPP/TB indices, along with increases in FCI and FML mentioned in our study, were based on visual observations rather than statistical analyses. Each parameter was represented by a single data point, which did not allow for formal statistical analysis to support the significance of these changes. To avoid any potential misunderstandings, we have amended the text by removing the term “significant”.

Q11 L 457: The authors should specify what type of ranching is being considered in the area. What is the target organism for which sea ranching has been implemented in the area?

Response: The Jinggong Marine Ranching is classified as a “recreational” type according to the classification standards of the Ministry of Agriculture and Rural Affairs. The targeted species for conservation and enhancement include the Yellowfin Seabream (Acanthopagrus latus), False Kelpfish (Sebastiscus marmoratus), Japanese Prawn (Marsupenaeus japonicus), Japanese Stone Crab (Charybdis japonica), Sea cucumber (Stichopus variegatus), Pacific Oyster (Crassostrea gigas), and Small Giant Clam (Lutraria sieboldii). Given that these sentences primarily serve as an introduction to Jinggong Marine Ranching, we have positioned this information in the “Materials and Methods, Introduction of the marine ranching” section of the manuscript.

Q12 L 461-462: The authors should explain more clearly how the use of artificial reefs affects nutrient dynamics within the benthic environment. They should discuss the biochemical processes involved and their ecological implications.

Response: According to your suggestion, we have expanded our discussion on the impact of artificial reefs on nutrient dynamics within the benthic environment. We have also discussed the biogeochemical cycling processes related to these nutrient dynamics and their ultimate ecological effects in the manuscript. Please refer to the relevant sections for more information.

Q13 L 486: In response to the reviewer’s comment, we have discussed approaches to improve energy transfer efficiency (TE) within the ecosystem and provided specific recommendations, as outlined below:

Optimizing the food web to facilitate smoother energy transfer pathways and increasing the biomass of higher trophic level (TL) organisms (such as those at TLs III and IV) are two essential strategies for improving TE values across TLs II to V. Mid-TL organisms, including crustaceans, cephalopods, and bivalves, play a pivotal role as intermediaries in the energy transfer between primary producers and apex predators. Therefore, we propose employing strategies such as habitat restoration, biological conservation, and stock enhancement to increase the diversity and biomass of mid-TL organisms. These efforts will help optimize the food web structure and enhance the biomass of higher TL organisms, thereby improving energy transfer efficiency (TE) across TLs II to V.

Q14 L 494: The authors should be more explicit about how external disturbances disrupt the self-organisation processes within the study area. For example, they should consider climate change or pollution and their effects on ecosystem dynamics.

Response: According to your suggestion, we have revised the manuscript to provide more clarity on how external disturbances, such as fishing activities and marine natural disasters, can disrupt the self-organization process of the ecosystem. Specifically, we have included an example of how massive waves and strong winds caused by typhoons can disturb seabed sediments, leading to significant decreases in the abundance and species composition of large benthic organisms in a short period. This disturbance, in turn, hinders the ecosystem's ability to achieve higher levels of organization.

Q15 L 503: It is unclear which species are overfished in the study area.

Response: In response to your comment, we have clarified in the manuscript that, based on the analysis presented in Figure A15, the biomass of large crabs is predicted to decline to zero by around 2027, under the current fishing levels. Therefore, we conclude that large crabs are the primary species that are currently overfished in the ecosystem of the study area.

Q16 L 527-528: The statement 'While zooplankton can quickly respond to ecosystem disturbances, bivalves exhibit slower responses, thus creating a pathway for slow energy flow' is controversial, as mussels can also respond quickly to disturbances.

Response: What we mean by “quickly respond to ecosystem disturbances” is relative to the turnover time of the organisms. For example, in this study, the annual production-to-biomass (P/B) ratio of zooplankton is 32.54/year, and its bimass turnover time is only 11.21 days, whereas the annual P/B ratio of mussels is 5.06/year, and its biomass turnover time is 73 days. Therefore, when external disturbances occur, zooplankton can respond quickly through rapid reproduction and growth, with their population size quickly declining under adverse environmental conditions due to high mortality. However, when the external environment improves, zooplankton can rapidly rebuild, thus facilitating energy transfer to higher trophic levels. On the other hand, when mussels face external disturbances, although they also respond quickly in terms of physiology, the speed of their population changes is slower compared to that of zooplankton. As mussel populations decline, they can maintain a relatively stable population for a longer period, thus transferring energy to higher trophic levels. When environmental conditions improve, however, the population recovery of mussels is notably slower than that of zooplankton due to the time required for mussel growth and reproduction, which leads to a slower rate of energy transfer to higher trophic levels compared to zooplankton. As Rooney et al. (2022) stated, “fast energy channels tend to have smaller, faster growing populations that have higher biomass turnover rates compared with the slow energy channels. When these systems are perturbed, populations in the fast channel respond quickly, allowing for the rapid recovery of predator populations.”

References

Rooney N, Mccann KS. Integrating food web diversity, structure and stability. Trends in Ecology & Evolution, 2012, 27(1):40-46. DOI:10.1016/j.tree.2011.09.001.
